# Rescue of mitochondrial import failure by intercellular organellar transfer

Hope I. Needs [1], Emily Glover[1], Gonçalo C. Pereira[1,3], Alina Witt[2], Wolfgang Hübner [2], Mark P. Dodding [1], Jeremy M. Henley [1] & Ian Collinson [1]

Mitochondria are the powerhouses of eukaryotic cells, composed mostly of nuclear-encoded proteins imported from the cytosol. Thus, problems with the import machinery will disrupt their regenerative capacity and the cell's energy supplies – particularly troublesome for energy-demanding cells of nervous tissue and muscle. Unsurprisingly then, import breakdown is implicated in disease. Here, we explore the consequences of import failure in mammalian cells; wherein, blocking the import machinery impacts mitochondrial ultra-structure and dynamics, but, surprisingly, does not affect import. Our data are consistent with a response involving intercellular mitochondrial transport via tunnelling nanotubes to import healthy mitochondria and jettison those with blocked import sites. These observations support the existence of a widespread mechanism for the rescue of mitochondrial dysfunction.

Mitochondria are semi-autonomous organelles, found inside eukaryotic cells, responsible for oxidative phosphorylation (OXPHOS) and metabolic control, amongst other regulatory and biosynthetic functions. Their small genome encodes only a few proteins (13 in humans) while the remaining proteome (>1000 proteins) is synthesised in the cytosol and imported via highly specialised protein translocation machinery. The major route is the presequence pathway, taken by precursor proteins usually (-70%) with cleavable N-terminal mitochondrial targeting sequences (MTS)[1]. These proteins are recognised at the surface of the outer mitochondrial membrane (OMM) by receptors of the translocase of the outer membrane (TOM) complex. Transport then proceeds through the TOM40 channel into the intermembrane space, towards the translocase of the inner membrane (TIM23) complex[2,3]. Additional internal sorting sequences are also deployed, which target the translocating precursor through the TIM23[SORT] complex to be inserted laterally into the inner mitochondrial membrane (IMM)[2]. Otherwise, transport proceeds across the IMM and into the matrix via the TIM23[MOTOR] complex. Passage of proteins into and across the IMM both require the membrane potential (ΔΨ), while pulling the protein into the matrix also requires ATP turnover[2]. MTS cleavage then enables the release and folding of the mature protein;

and, where required, assembly into multi-subunit complexes. A recent kinetic analysis shows that passage into the matrix occurs with two distinct rate-limiting steps:[3] (1) transport through TOM and (2) initiation at TIM23, followed by fast (non-rate-limiting) transport through TIM23; rather than by continuous passage through TOM-TIM23.

There are multiple potential causes of mitochondrial import failure. Age, exposure to environmental and endogenous (e.g., reactive oxygen species–ROS) toxins, as well as nuclear and mitochondrial mutations, can all impair bioenergetic function, resulting in depletion of ΔΨ and ATP required for import. Additionally, the TOM and TIM complexes can become blocked, e.g., by stalled/misfolded precursor proteins. Indeed, refined kinetic analyses of protein translocation through the mitochondrial import and bacterial secretion machinery[3–5] show that transport failure and blockages are integral components of the reaction cycle. Another possibility is that aggregation-prone proteins, such as those variants implicated in neurodegenerative disease (amyloid precursor protein (APP), alpha-synuclein, Huntingtin, and Tau) associate with the import machinery and perturb the transport process[6–9].

The consequences of mitochondrial import dysfunction, and thereby biogenic failure, are catastrophic for cellular health;

[1]School of Biochemistry, University of Bristol, Bristol BS8 1TD, UK. [2]Fakultät für Physik, Universität Bielefeld, Bielefeld Postfach 100131 D-33501, Germany. [3]Present address: Nanna Therapeutics, Merrifield Centre, Rosemary Lane, Cambridge CB1 3LQ, UK. ✉e-mail: J.M.Henley@bristol.ac.uk; ian.collinson@bristol.ac.uk

particularly for high energy-consuming cells such as those of muscle and nervous tissue. Indeed, there are strong links between mito-chondrial import failure and neurodegeneration, see ref. 10 and references therein. Deficient import leads to the build-up of cytosolic precursor proteins, inducing the inhibition of protein synthesis, acti-vation of the proteasome, and mitoprotein-induced stress responses[11–15]. While these responses have been shown to mitigate against the potentially toxic effects of large quantities of aggregation-prone precursor, they do not always successfully resolve the under-lying problem of cells incapable of mitochondrial regeneration. To maintain overall bioenergetic function, mitochondria with failing import sites either need to be rectified or replaced.

This paper describes a rescue mechanism in cells subject to per-turbation of the mitochondrial import machinery that is consistent with the formation of tunnelling nanotubes (TNTs), enabling inter-cellular mitochondrial transport—potentially for the recruitment of viable mitochondria from healthy to compromised cells, replacing defective mitochondria moving in the other direction.

## Results

### Precursor stalling affects mitochondrial morphology

To study the consequences of import failure, and the response, we established tools to monitor and artificially block mitochondrial pro-tein import in intact cells. HeLa cells were conditioned in galactose (HeLaGAL) to maximise their dependency on mitochondrial OXPHOS, in contrast to those grown in glucose (HeLaGLU), which favour gly-colytic metabolism[16]. These cells were then used to study the impact of precursor stalling. Precursor proteins are imported in an unfolded state, so the C-terminal fusion of dihydrofolate reductase (DHFR), which resists unfolding when bound to the drug methotrexate (MTX), can be exploited to block the import machinery (Fig. 1a)[17,18]. We, therefore, engineered a precursor construct encoding the MTS of the ATP synthase Subunit 9 (Su9) fused to a fluorescent reporter protein (EGFP or mScarlet) and DHFR for over-production in HeLaGAL cells. Confocal microscopy confirmed that the mature protein was suc-cessfully targeted to mitochondria in the absence of MTX, but to a lesser extent in its presence (Fig. 1b); note, characterisation of import

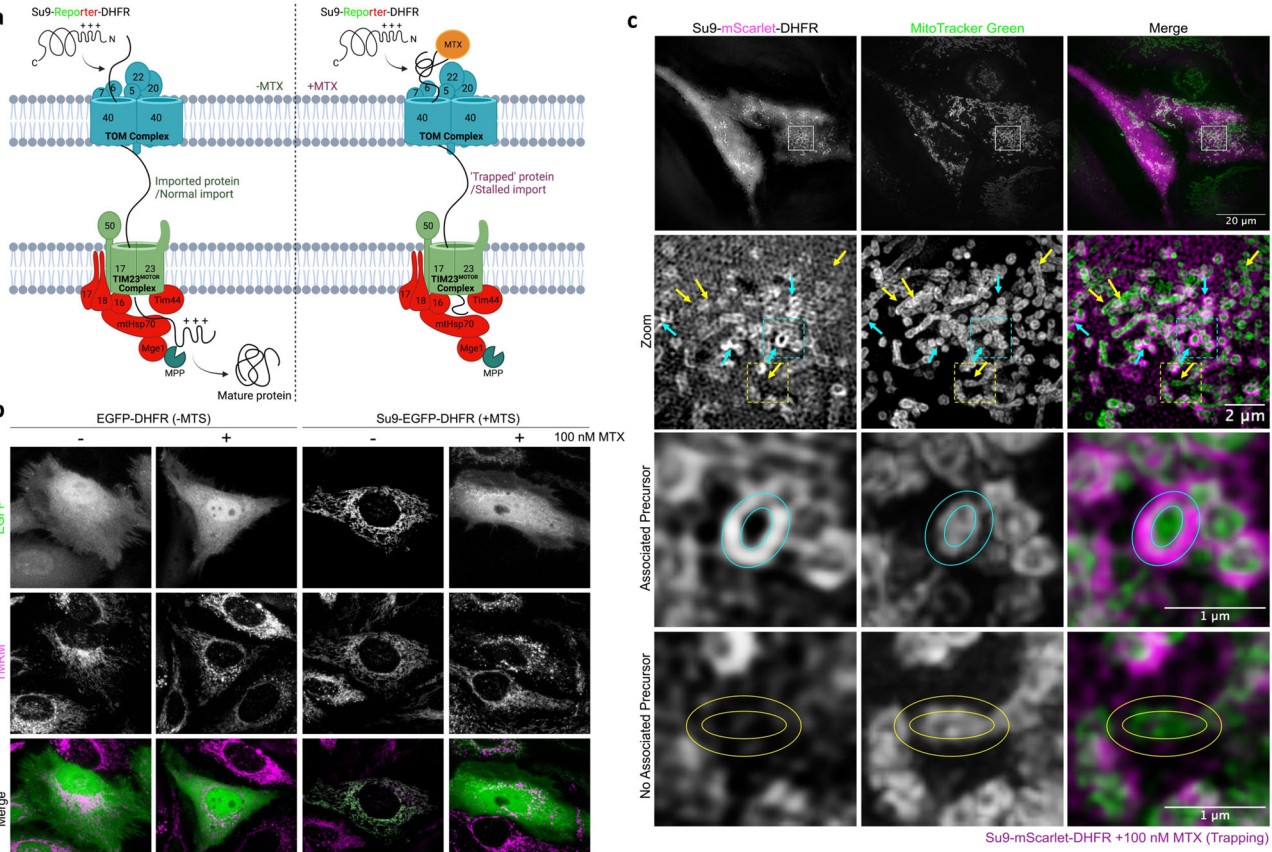

**Fig. 1 | Schematic overview and demonstration of DHFR-MTX precursor trap-ping system in mammalian cells. a** In the absence of MTX (*left*), the precursor protein (Su9-Reporter-DHFR) is imported into mitochondria via the presequence pathway. In the mitochondrial matrix, the MTS is cleaved by MPP, and it folds to form the mature protein. In the presence of MTX (*right*), MTX binds to DHFR in the cytosol, preventing it from unfolding, and thus from crossing the TOM40 channel. Since the protein contains a long Su9-Reporter region before the DHFR, this region forms a plug through the TOM40 and TIM23 channels, acting as a stalled or aggregated precursor protein within the presequence pathway. Note that the MTS of sufficiently long trapped precursors could reach the matrix, and thereby also be subject to MPP processing. Schematic created using BioRender. **b** Representa-tive confocal images of HeLaGAL cells over-producing EGFP-DHFR (- MTS; green) or Su9-EGFP-DHFR (+ MTS; green) with or without 100 nM MTX for 48 h.

Mitochondria were stained with 25 nM TMRM (magenta). **c** Representative SR 3D-SIM images showing trapping substrate accumulation around mitochondrial membranes (MitoTracker Green, green) in HeLaGAL cells over-producing Su9-mScarlet-DHFR (magenta; 48 h) in the presence of 100 nM MTX (48 h). Cyan arrows indicate precursor-mitochondria surface association (zoom in cyan box), yellow arrows indicate 'naked' mitochondria without surface associated precursor (zoom in yellow box). Circles indicate the area within which pixel intensity was analysed to determine precursor-mitochondria association. *N* = 3 biological replicates (**b**, **c**). MPP mitochondrial processing peptidase, mtHsp70 mitochondrial heat shock protein 70, MTS mitochondrial targeting sequence, MTX methotrexate, SR 3D-SIM super-resolution three-dimensional structured illumination microscopy, TIM translocase of the inner mitochondrial membrane, TOM translocase of the outer mitochondrial membrane, TMRM tetramethylrhodamine methyl ester.

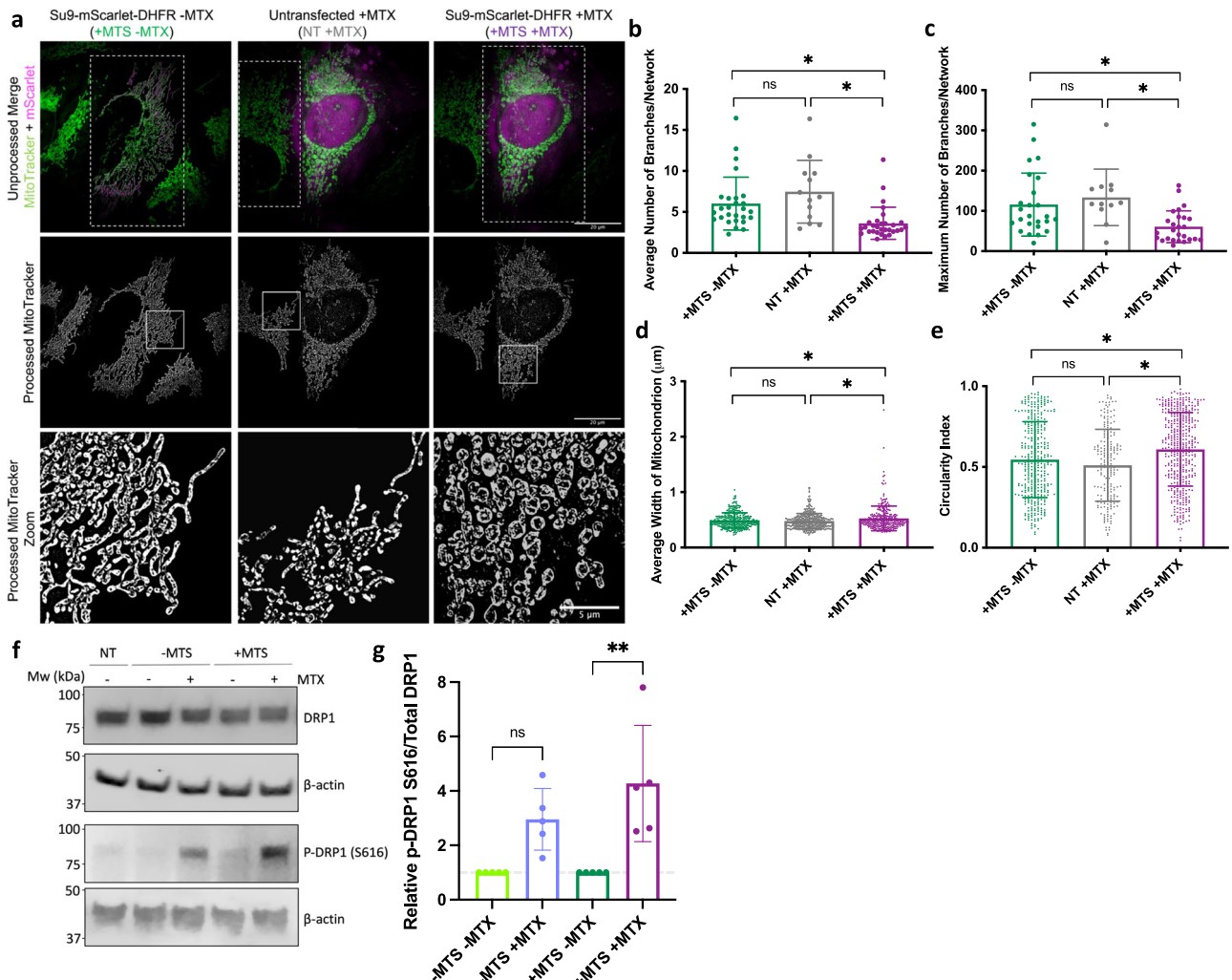

**Fig. 2 | Chronic precursor trapping alters mitochondrial morphology and dynamics. a** Representative SR 3D-SIM images showing mitochondria in HeLaGAL cells subjected to trapping (+ MTS + MTX; 48 h) or not (+ MTS - MTX). Not transfected (NT) cells in the same well as cells subjected to trapping were used as an internal control for MTX treatment (NT + MTX; 48 h). The top row shows a merge of MitoTracker Green (mitochondria; green) and mScarlet (trapping precursor; magenta) prior to image processing. The middle row shows mitochondria after image processing. Box shows zoom area. N = 3 biological replicates. **b–e** Quantification of mitochondrial morphology. For *branching* (**b, c**), each point represents an individual cell from a separate field of view. n = 27, 13, 30 cells for different conditions, respectively. P = 0.8181, 0.0323, 0.0422 (**b**), P = 0.8041, 0.0380, 0.0458 (**c**). For *width* (**d**), each point represents an individual mitochondrion, as an average of 5 measurements per mitochondrion from a region from a selected field of view (independent cell). n = 360, 470, 315 mitochondria, from 20 cells for each condition, respectively. P = 07661, 0.0332, 0.0158. For *circularity* (**e**), each point represents an individual mitochondrion. n = 359, 199, 434 mitochondria,

respectively. P = 0.3502, 0.0117, 0.0447. **f** Representative Western blots showing the abundance of total DRP1 and phospho-DRP1 S616 in HeLaGAL cells overproducing EGFP-DHFR (- MTS) or Su9-EGFP-DHFR (+ MTS), or untransduced (NT) cells in the absence or presence of 100 nM MTX (48 h). β-actin was used as a loading control. N = 5 biological replicates. **g** Quantification of (**f**), normalised to β-actin loading control and respective ±MTS control. N = 5 biological replicates represented by individual data points. P = 0.0881, 0.0029. Statistical significance was determined using nested (**b–e**) or normal (**g**) one-way ANOVA and Tukey's tests. Data are presented as mean values ± S.D., and P values are reported left-right, bottom-top (**b–e, g**). Source data and uncropped blots are provided in the Source Data file (**b–e, g**). DRP1 dynamin related protein 1, kDa kilodalton, MPP mitochondrial processing peptidase, mtHsp70 mitochondrial heat shock protein 70, MTS mitochondrial targeting sequence, MTX methotrexate, Mw molecular weight, NT not tranfected, p-DRP1 S616 DRP1 phosphorylated at Serine 616, SR 3D-SIM super-resolution three-dimensional structured illumination microscopy.

failure in the control without an MTS. Interestingly, in the cells subject to trapping, structured illumination microscopy (SIM) showed that some mitochondria (an average of 43.3% (±39.4)) were enveloped by precursor protein, while others were not (Fig. 1c – cyan and yellow arrows, respectively; Supplementary Figs. 1a–c and 2.).

High-resolution SIM also revealed that artificial precursor trapping elicited a change in mitochondrial morphology (Fig. 2a–e). The mitochondria became less branched, more rounded, and wider; indicative of fragmentation. These effects did not occur when the precursor construct was not produced (in untransfected neighbouring cells), so they were not due to off-target effects due to exposure to MTX (Fig. 2a–e – NT + MTX).

We next analysed the abundance of a known regulator of mitochondrial fission, DRP1, which assembles into spiral-like oligomers when recruited to the OMM. Constriction brings about fission, mediated by GTP hydrolysis[19] and regulated by various post-translational modifications, including phosphorylation of DRP1 at serine-616 (p-DRP1 S616)[20]. Western blotting showed that the abundance of total DRP1 was unchanged in HeLaGAL cells following mitochondrial precursor stalling, but phosphorylation at the pro-fission site S616 was increased (Fig. 2f, g). While there is an effect of MTX alone, the increase in the ratio of p-DRP1 S616/ total DRP1 in the - MTS + MTX condition was not significant. This effect was greater and more significant following precursor trapping (+ MTS + MTX). This noted increase in

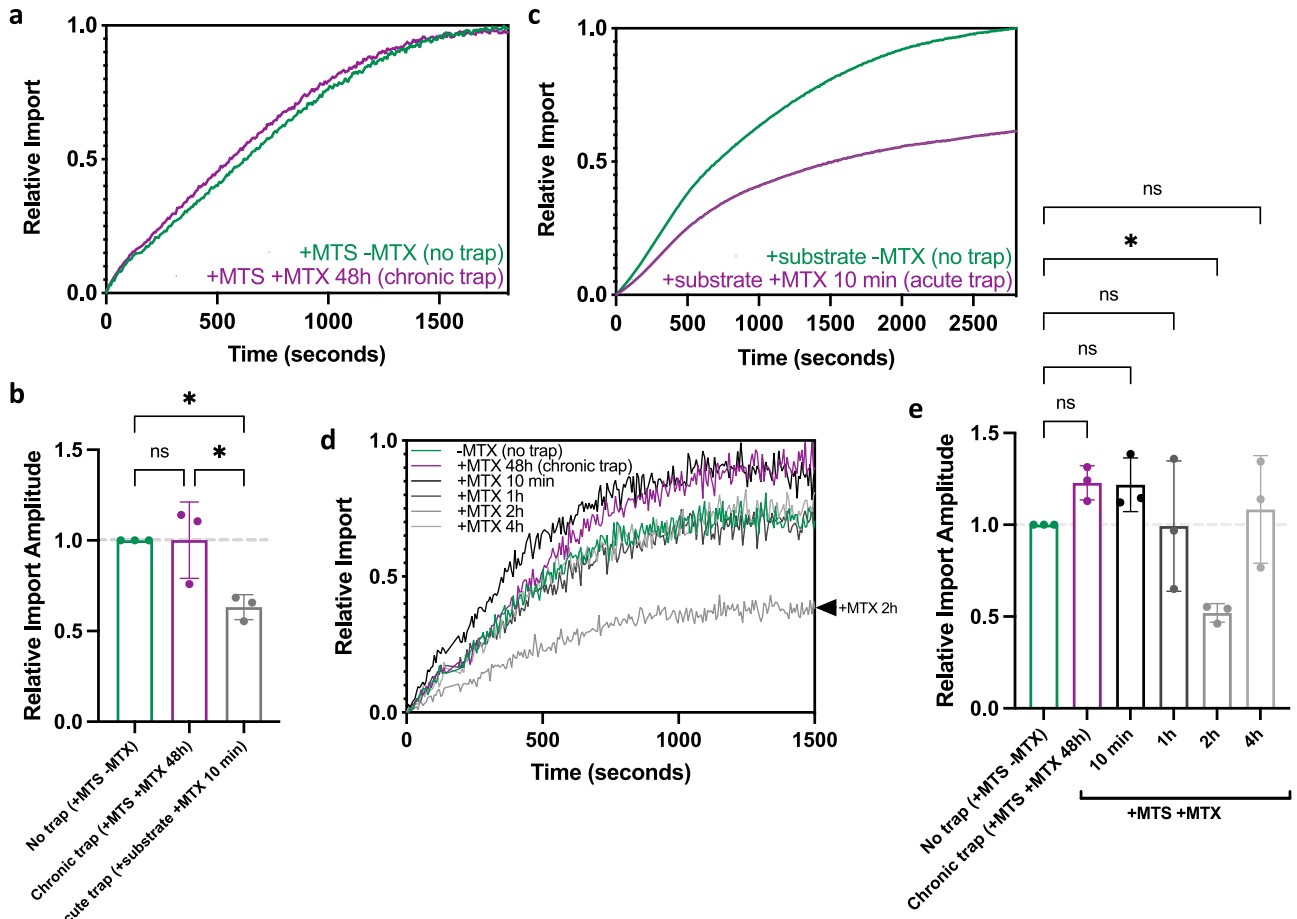

**Fig. 3 | Acute but not chronic precursor trapping impacts import yield.**
**a** MitoLuc import trace following chronic trapping (48 h; purple), or no trapping (precursor imported; green). **b** Import amplitude following chronic trapping (48 h; purple), acute trapping (10 min; grey), or no trapping (precursor imported; green). $P = 0.9999, 0.0283, 0.0289$. **c** MitoLuc import trace following acute trapping (10 min; grey), or no trapping (precursor imported; green). MitoLuc import trace (**d**) and amplitudes (**e**) following in-cell precursor trapping for 48 h (purple; chronic trap), 10 min, 1 h, 2 h, or 4 h (grey; trap time course), or no trapping (precursor

imported; green). $P = 0.5263, 0.5672, >0.9999, 0.0488, 0.9792$. $N = 3$ biological replicates, each with $n = 3$ technical replicates in HeLaGAL cells (**a**–**e**). Statistical significance was determined using one-way ANOVA and Tukey's tests and $P$ values are reported left-right, bottom-top (**b**, **e**). Data are presented as mean values ± S.D. with individual data points for each biological replicate (**b**, **e**) or as representative traces (**a**, **c**, **d**). Raw data are provided in the Source Data file (**a**–**e**). MTS mitochondrial targeting sequence, MTX methotrexate.

p-DRP1 S616 may be related to mitochondrial import dysfunction induced by precursor trapping, though the biological significance of this is unclear from the data presented. We extended our analysis to other markers of mitochondrial dynamics, including OPA1, the processing of which regulates mitochondrial dynamics, as well as MFF and FIS1, both of which function as DRP1 receptors and play crucial roles in mitochondrial fragmentation. We observed no changes in the processing of OPA1 or the abundance of MFF or FIS1 upon precursor trapping beyond the off-target effects of MTX treatment (Supplementary Fig. 3a, b).

### Chronic trapping fails to inhibit mitochondrial import

We previously developed a split luciferase assay to monitor protein import into isolated yeast mitochondria[21], which we successfully implemented and validated within HeLa cells[22]. Briefly, HeLa cells were cultured with the large fragment of the luciferase (11S) segregated in the mitochondrial matrix, so that the import of a precursor with the small fragment (pep86–required for luminescence) fused to the C-terminus could be monitored. This capability enabled the reporting of the import of an exogenously added precursor in plasma membrane permeabilised HeLaGAL cells that had been previously subject to precursor stalling by DHFR-MTX.

We first evaluated import dynamics after chronic exposure to precursor stalling (48 h of Su9-mScarlet-DHFR over-production + MTX; see above). Surprisingly, we observed no effect on the import of a second precursor protein Su9-EGFP-pep86 in the in-cell (HeLaGAL) assay (Fig. 3a, b). However, under more acute stalling conditions–achieved by the introduction of high concentrations of purified precursor protein-DHFR fusion, MTX and reduced nicotinamide adenine dinucleotide phosphate (NADPH; for increased stabilisation of DHFR) for 10 min–the import of the reporter precursor diminished (Fig. 3b, c; acute trap).

To ensure these results were not attributable to the artificial nature of the acute endogenous trapping set-up, we conducted a time course of trapping in the in-cell system by transfecting cells with the trapping construct and treating them with MTX for 48 h (chronic trap), 4 h, 2 h, 1 h, and 10 min. After 2 h of trapping, we observed a reduction in import amplitude of 48.1% (±5.1), which was subsequently lost from 4 h and beyond up until 48 h (Fig. 3d, e).

Since analogous treatments of isolated mitochondria in vitro completely blocked the import sites[3], the preservation of mitochondrial import inside cells following chronic precursor trapping (4–48 h) was puzzling. In line with this, mitochondrial stress tests measuring oxygen consumption and membrane potential

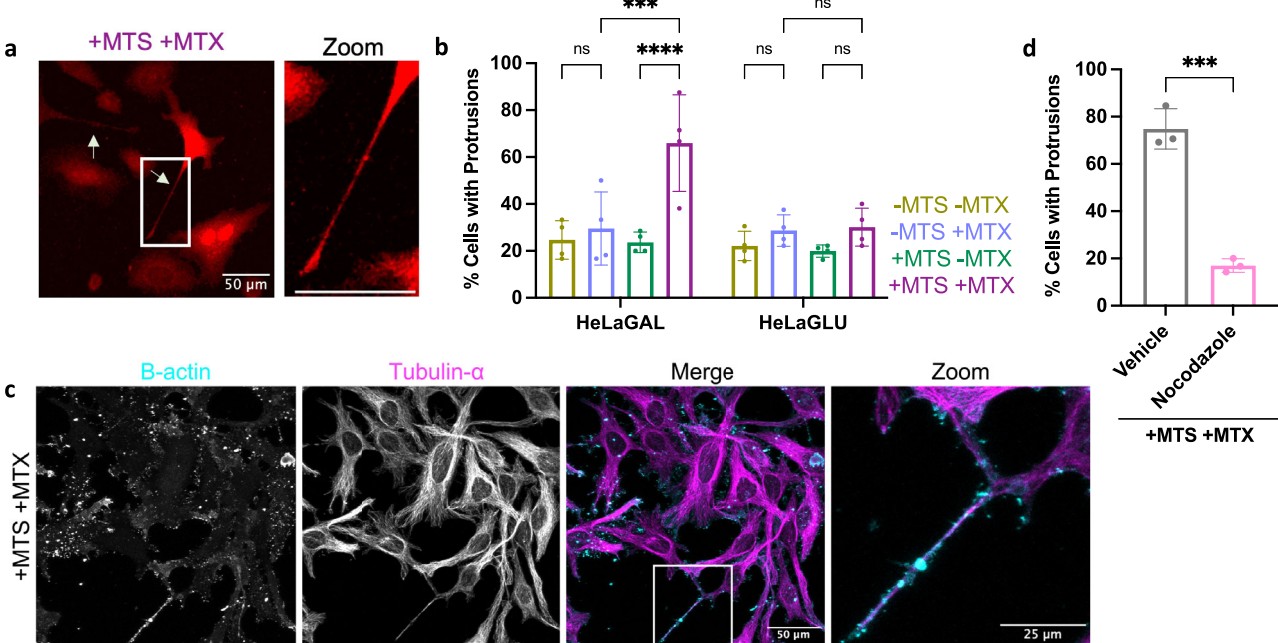

**Fig. 4 | Chronic precursor trapping induces the formation of TNTs.**
**a** Representative confocal images showing HeLaGAL cells over-producing mCherry (red) and Su9-EGFP-DHFR in the presence of 100 nM MTX for 48 h (+ MTS + MTX; chronic trapping). Arrows indicate protrusions and the box highlights zoom region. **b** Quantification of (**a**) demonstrating the proportion of transfected cells with protrusions. N = 4 biological replicates; 20 cells counted per replicate. P = 0.9886, <0.0001, 0.9505, 0.7149, 0.0004, >0.9999. **c** Representative confocal images showing tubulin-α and β-actin staining of protrusions following chronic trapping (48 h) in HeLaGAL cells. N = 4 biological replicates. **d** Quantification of the

proportion of HeLaGAL cells, subjected to chronic precursor trapping (48 h), with protrusions following incubation for 48 h in the absence (vehicle; DMSO only) or presence of 100 nM nocodazole (48 h). N = 3 biological replicates; 20 cells counted per replicate. P = 0.0004. Statistical significance was determined using a two-way ANOVA and Šidák's test (**b**) or a two-tailed, unpaired t test (**d**). Data are presented as mean values ± S.D. with individual data points for each biological replicate and P values are reported left-right, bottom-top (**b**, **d**). Raw data are provided in the Source Data file (**b**, **d**). MTS mitochondrial targeting sequence, MTX methotrexate, TNT tunnelling nanotube.

(tetramethylrhodamine methyl ester (TMRM) fluorescence) showed that, despite trapping-induced changes in mitochondrial morphology, the membrane potential and respiratory capacity were unaffected by precursor stalling (Supplementary Fig. 4a–c). Based on these findings, we hypothesised that a clearance or rescue mechanism must be operating; one that is lost following mitochondrial isolation.

## Chronic trapping induces protrusions consistent with tunnelling nanotubes

We observed that the cells subject to chronic exposure to precursors with DHFR and MTX (chronic trapping), exhibited an increase in thin elongated protrusions (Fig. 4a, b; Supplementary Fig. 5). The increase in protrusions was dependent on DHFR-induced precursor stalling, and not an off-target effect of MTX, because they were less apparent when the MTS was omitted. Furthermore, the trapping-induced increase in the proportion of cells with protrusions only occurred when HeLa cells were cultured in galactose media, and not when they were grown in glucose media (Fig. 4b; Supplementary Fig. 5). When HeLa cells are cultured in a galactose-based medium, glutamine accounts for ~98% of ATP production, which forces cells into OXPHOS[16]. This causes cells to become reliant on their mitochondria in a similar manner to those of brain or muscular tissue. Moreover, galactose-cultured cells have previously been shown to reveal mitochondrial dysfunction whereas glucose-cultured cells do not[16].

The fact that this occurrence was dependent on galactose conditioning is consistent with a response mediated by mitochondria. Immunocytochemistry and confocal microscopy showed that the protrusions contain microtubules and actin (Fig. 4c), characteristic of TNTs, which are capable of organellar (including mitochondrial) transport between cells[23–26]. To further investigate the identity of these protrusions, cells were treated with nocodazole, which impairs TNT

establishment by inhibition of microtubule formation[27–31]. In doing so, the number of protrusions was reduced to the baseline levels exhibited by cells in the absence of an MTS and MTX (- MTS - MTX; Fig. 4b, d; Supplementary Fig. 6), consistent with the possibility that they are TNTs

## Intercellular mitochondrial transport might occur by tunnelling nanotubes

We next investigated whether these possible TNTs were capable of intercellular mitochondrial transport to recruit functional mitochondria or remove those that have become compromised. Close inspection by super-resolution imaging highlighted the presence of mitochondria inside tubes connecting a cell challenged by precursor stalling with an untransfected healthy cell (Fig. 5a). To confirm mitochondria were transferred between cells, co-cultures of challenged HeLaGAL cells (producing a stalling precursor and a green mitochondrial marker: Su9-EGFP) and healthy cells (producing a red mitochondrial marker protein: Cox8a-DsRed) were established in the presence of MTX and imaged by confocal microscopy.

The images identified cells containing a mixture of mitochondria originating from both healthy and challenged cells (Fig. 5b). We performed correlation analysis to quantify the degree of colocalisation between mitochondrial markers in cells with mitochondrial mixing compared to internal controls (Supplementary Fig. 7). The data confirmed that, following mitochondrial mixing, mitochondrial markers were colocalised, indicating that mitochondria were transferred and successfully integrate into the mitochondrial network of the recipient cell (Supplementary Fig. 7). Furthermore, live cell light-sheet fluorescence microscopy showed healthy mitochondria (without precursor stalling) being transferred through a putative TNT towards a challenged cell (Fig. 5c; Supplementary Movie 1). This observation explains

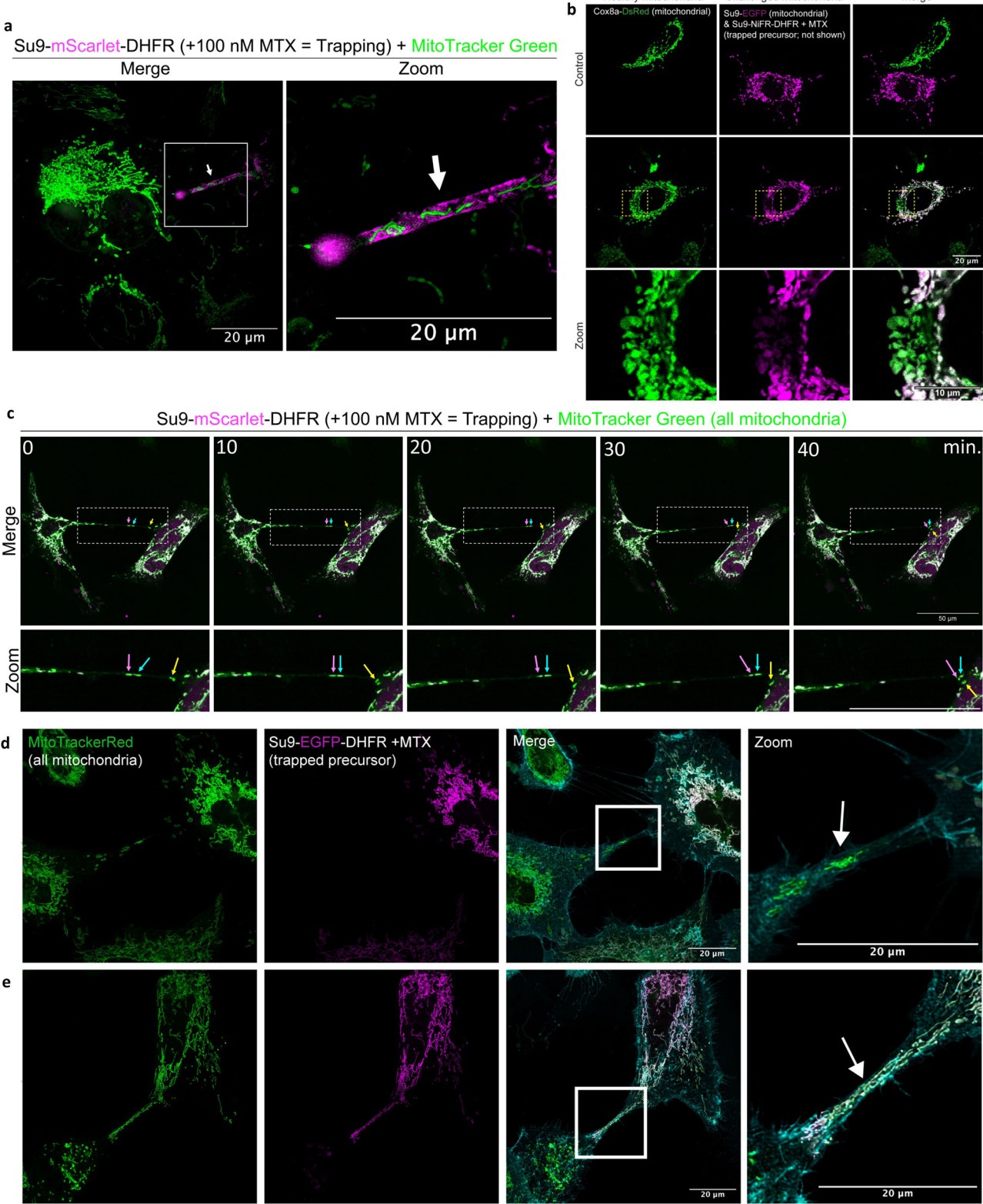

why, within a cell exposed to precursor trapping, some mitochondria are surrounded by aggregated precursors and thereby stalled, while others in the same cell are not (Fig. 1c; Supplementary Figs. 1a–c and 2). The fact that we identify cells with mixed mitochondrial populations (having undergone mitochondrial rescue) after only 8 h of co-culture (Fig. 5b), and observe mitochondria being transferred via putative TNTs up to 48 h of trapping (Fig. 5c), suggests that the rescue response is both fast and continuous.

Further analysis by confocal microscopy identified protrusions containing healthy mitochondria, travelling from a healthy cell towards a cell with trapped precursor (Fig. 5d). Conversely, there were other instances where protrusions were observed containing mitochondria originating from cells with trapped precursor (Fig. 5e). Thus, it appears that mitochondria travel in both directions; potentially for replenishment from healthy cells, and for affected cells to jettison impaired mitochondria. Interestingly, while mitochondria with

**Fig. 5 | Chronic precursor trapping induces intercellular mitochondrial transfer via TNTs. a** Representative SR 3D-SIM image showing mitochondria (Mito-Tracker Green; green) within a TNT formed between a HeLaGAL cell with trapped precursor (48 h; Su9-mScarlet-DHFR; magenta), and an untransfected neighbouring cell (green mitochondria, unlabelled cytosol). The arrow highlights the TNT, and the box indicates the zoom area. **b** Representative confocal images from co-culture experiments, showing mitochondrial transfer between healthy cells (healthy mitochondria labelled with Cox8a-DsRed; green) and cells exposed to precursor trapping (challenged mitochondria labelled with Su9-EGFP; magenta). The middle panel (transfer) shows a cell where mitochondrial transfer has occurred, as demonstrated by the colocalization of the two mitochondrial markers, also shown in zoom indicated by box. Internal control (no transfer; top panel) shows neighbouring cell in co-culture with no mitochondrial transfer. **c** Time series of still images from a representative time-lapse movie showing the transfer of healthy mitochondria (MitoTracker Green; green) within a TNT from a healthy cell (left) into a cell with trapped precursor (Su9-mScarlet-DHFR (faint cytosolic staining in magenta); 48 h; right). **d** Representative confocal images showing mitochondria (MitoTracker Red; green) within a TNT (plasma membrane stained with WGA; cyan) being transferred from an untransfected, healthy HeLaGAL cell (left; green mitochondria; no trap) to a neighbouring cell with trapped precursor (right; 48 h; Su9-EGFP-DHFR; magenta) or **e** in the other direction, from a cell with trapped precursor (right) to a healthy neighbouring cell (left). Box indicates the zoom area, and the arrow highlights the TNT. $N = 3$ biological replicates (**a**–**e**). MTS mitochondrial targeting sequence, MTX methotrexate, SR 3D-SIM super-resolution three-dimensional structured illumination microscopy, TNT tunnelling nanotube, WGA Wheat Germ Agglutinin.

trapped precursors are subject to transport, the cytosolic precursor proteins accumulated in the cytosol are not—suggesting that transport occurs actively, rather than passively by diffusion.

## Protrusion formation is a general response to import perturbation

To explore whether this rescue process is a general response to perturbation of the protein import apparatus, HeLa cells were exposed to MitoBloCK-20 (MB20)—a small molecule inhibitor of the presequence pathway acting on TIM17—a key component of the TIM23 complex[32]. Like precursor stalling, only acute exposure to the drug affected protein import activity (Fig. 6a, b). The failure of chronic drug exposure to impact protein import coincides with a substantial increase in the proportion of cells with structures resembling TNTs (Fig. 6c, Supplementary Fig. 8); once again this only occurred in cells highly dependent on mitochondria for ATP synthesis (HeLaGAL). These putative TNTs, induced by chemical inhibition, also contained actin, microtubules, and mitochondria (Supplementary Fig. 9).

The results presented thus far suggest that the protrusions are part of a widespread rescue mechanism that enables cells containing mitochondria with compromised import to receive functional mitochondria from nearby healthy cells and unload dysfunctional mitochondria for degradation/regeneration. To test this hypothesis, we re-examined protein import, which was apparently unaffected in cells subjected to chronic (48 h) precursor stalling (Fig. 3a, b), but in the presence of the TNT inhibitor nocodazole. In cells exposed to chronic precursor trapping and treated with nocodazole, we observed a reduction of 33.5% (±7.0) in the total amount of protein imported compared to cells without trapped precursor (Fig. 6d, e). Additionally, this analysis established that nocodazole does not affect the mitochondrial import activity of cells in the absence of precursor trapping (Fig. 6d, e).

The amplitudes reported by the MitoLuc assay reflect the quantities of precursor entry into the matrix, allowing us to approximate the proportional loss of import function. Chronic trapping in the presence of nocodazole (to destabilise TNTs) brings about a reduction in import of 33.5% (±7.0) (Fig. 6d, e); similar to the loss observed with acute trapping 36.8% (±6.9), achieved by the addition of purified trapping precursor (Fig. 3b, c). Before the rescue response is fully underway, after only 2 h of trapping (Fig. 3d, e), this reduction rises to 48.1% (±5.1). These values are likely to be suppressed by residual TNTs in the presence of nocodazole (~20%; Fig. 4d), underlying rescue during acute trapping, and/ or basal levels of a rescue process independent of import machinery perturbation or TNTs. The recovery of this lost half (or possibly higher) of import function is consistent with the observed high proportions of mitochondrial mixing (Fig. 5b, Supplementary Fig. 7).

The results of the trapping time course also provide us with information about the progression of the rescue mechanism. After 10 min and 1 h of trapping there was no significant impact on mitochondrial import function; presumably, at these early time points, the cells have yet to produce sufficient trap to have an effect. Only after 2 h do we see consequences for import (Fig. 3d, e). Subsequently, after 4 h of trapping, the import function returned to comparable levels as in the absence of MTX (no trap; Fig. 3d, e). This indicates that within 2 h, the cells sense the import defect and transfer mitochondria intercellularly to rescue import function; presumably, due to TNT formation. However, for technical reasons we were unable to verify the formation of TNTs at this early time point, most likely due to their fragility and transient nature.

Overall, the data presented here are consistent with the unexpected resilience of import function, e.g., within cells subject to precursor trapping, being dependent on TNTs. However, further experimentation is necessary to support this conclusion.

## Stalled precursors recruit factors involved in TNT formation

To investigate this rescue mechanism further, we deployed tandem mass tagging mass spectrometry (TMT-MS) to identify proteins associated with the trapped precursor. Cells were subject to stalling through the production of a Su9-EGFP-DHFR fusion (with and without the MTS and MTX as controls). Following a chronic (48 h) treatment, mitochondria from HeLaGAL cells were isolated and proteins were gently extracted with glyco-diosgenin amphiphile (GDN). Proteins interacting with the precursor were then isolated using a GFP trap, which effectively binds the precursor (Fig. 7a, b; Supplementary Fig. 10). Note that the trapping conditions (Supplementary Fig. 10; + MTX) resulted in a lower yield of recovered precursor in the mitochondrial extract and the pulldown because of the large reduction of mitochondrial import. These levels were adjusted to parity for the subsequent TMT-MS comparison of the captured precursor. The analysis of precursor associates shows a significant enhancement of proteins when derived from cells cultured in the presence of MTX, compared to those grown in its absence. The volcano plot (Fig. 7b; Supplementary Data) highlight proteins in complex with the stalled precursor (+ MTX), controlled against those associated with the imported form (- MTX).

Of the significant hits with ≥2-fold enhancement, cell adhesion protein Fibronectin is critical for TNT formation and activity[33], whilst Apoptosis-inducing factor mitochondria associated 2 (AIFM2) is related to mitochondrial import stress signalling[34], and is induced by p53 activity[35] which can promote TNT formation[36]. Interestingly, there is also an enhancement of calcium signalling proteins: Calcium/calmodulin-dependent protein kinase type II subunit delta and S100-A2. Calcium signalling via the $Ca^{2+}$/Wnt pathway has been shown to regulate TNTs previously[37]. Additionally, cytoskeleton-related proteins required for TNT formation, such as Microtubule-associated protein 4 and Cdc42 effector protein 1, were also enhanced[38–40].

Associated proteins also include several proteins involved in stress response pathways, particularly associated with the proteasome (e.g., E3 ubiquitin ligase TRIM25[41], Proteasome subunit β type 5[42], and Heat shock cognate 71 kDa protein[43,44]), suggesting that other mitoprotein-induced stress response pathways are also at play. A

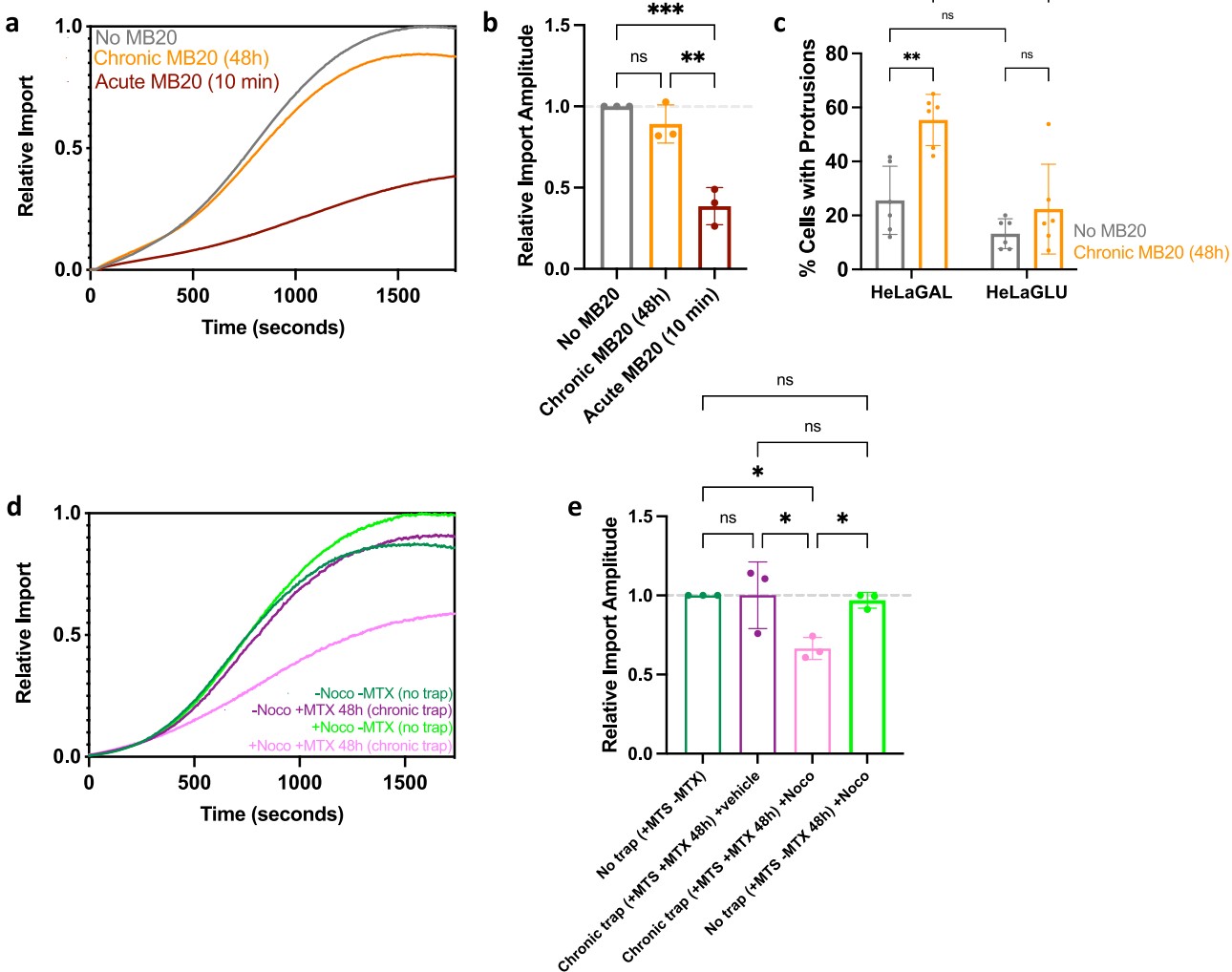

**Fig. 6 | Intercellular mitochondrial transfer via TNTs rescues import function.**
MitoLuc import trace (**a**) and amplitudes (**b**) following treatment with vehicle
(DMSO; grey), chronic (48 h; orange), or acute (10 min; red) MB20. *N* = 3 biological
replicates, each with 3 technical replicates. *P* = 0.3985, 0.0015, 0.0005.
**c** Quantification of the proportion of cells with TNTs following chronic treatment
with 10 μM MB20 (48 h). *N* = 6 biological replicates; 20 cells counted per replicate.
*P* = 0.0015, 0.5479, 0.2957, 0. 0005. MitoLuc import trace (**d**) and amplitudes (**e**) in
HeLaGAL cells subjected to no or chronic trapping (48 h) with vehicle (DMSO) or

100 nM nocodozole (48 h). *N* = 3 biological replicates. *P* = > 0.9999, 0.0276,
0.0448, 0.0283, 0.9845, 0.9867. Statistical significance was determined using one-
way (**b**, **e**) or two-way (**c**) ANOVAs with Tukey's tests. Data are presented as mean
values ± S.D. with individual data points for each biological replicate, and *P* values
are reported left-right bottom-top (**b**, **c**, **e**), or representative traces are shown
(**a**, **d**). Raw data are provided in the Source Data file (**a–e**). MB20 MitoBloCK−20,
MTS mitochondrial targeting sequence, MTX methotrexate, Noco nocodazole,
TNT tunnelling nanotube.

control experiment was also conducted without an MTS to show the
non-specific effects of MTX (Supplementary Fig. 11). In this control
there was only one significant enhancement in the presence of MTX,
Transportin-1, which was not significantly enhanced in + MTS + MTX.
Therefore, the hits identified as significantly enhanced represent true
associations with the trapped precursor.

To gain further insights into the impact of import failure on cel-
lular pathways, and their potential association with the rescue
mechanism described here, we analysed the significantly enhanced
proteins bound to the trapped precursor using the PANTHER classifi-
cation system tool. This analysis identified possible changes in several
cellular pathways, notably including cytoskeletal motor activity and
ATP-dependent activity (Supplementary Fig. 12; Supplementary
Table 1). Moreover, we investigated the proportion of associated
proteins that are mitochondrial versus non-mitochondrial, using
the MitoCoP compendium as a reference[45]. This highlighted that 21.3%
(55/238) of proteins associated with the trapped precursor were

mitochondrial, while the remaining 78.7% (203/238) were non-
mitochondrial (Supplementary Table 1 and Supplementary Data).

## Discussion

Taken together, our results demonstrate that impaired mitochondrial
import, be it precursor stalling at the outer membrane or inhibition of
the TIM23 complex, invokes a rescue mechanism involving the inter-
cellular transport of mitochondria via filamentous, membranous pro-
trusions containing F-actin and microtubules, consistent with their
identity as TNTs. TNTs have been described previously for their
involvement in long-range intercellular communication and transport
of lipids, nucleic acids, microRNAs, ions such as calcium, fragmented
plasma membrane, viruses (e.g., coronaviruses), and even entire
organelles: mitochondria, ER, Golgi, and endosomes[23–26,46–48]. The
dimensions of TNTs seem to be tailored according to the cargo, with
distinct thin and thick varieties, the latter with a diameter of 1−7 μm,
such as those seen here for mitochondrial transport[49–51]. TNT

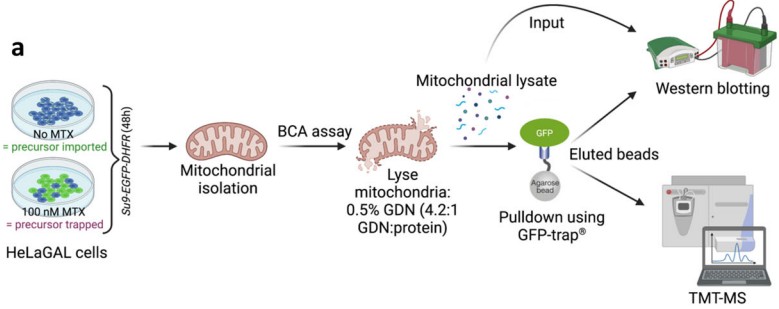

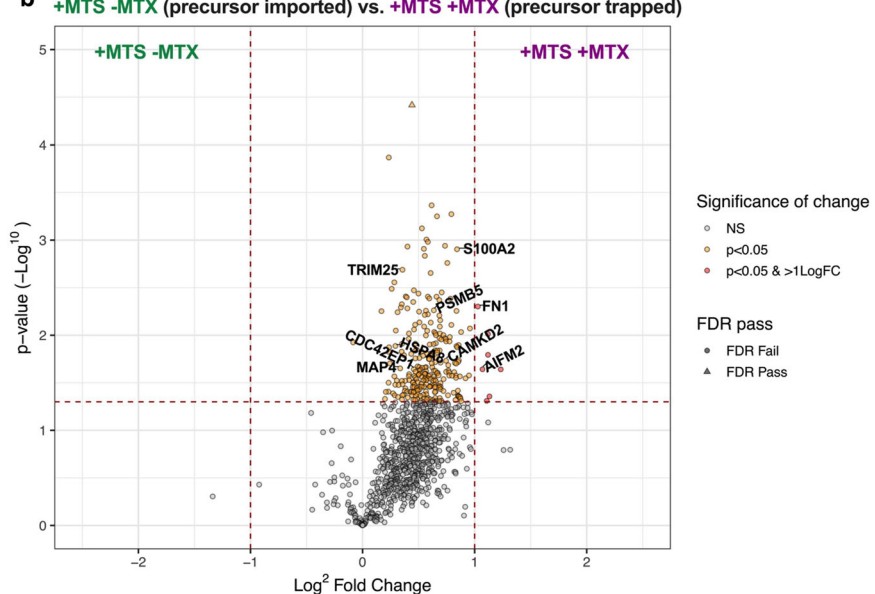

**Fig. 7 | TMT-MS highlights TNT-associated proteins enriched with trapped precursor. a** Schematic showing TMT-MS experimental outline. HeLaGAL cells were subjected to Su9-EGFP-DHFR over-production in the presence or absence of 100 nM MTX for 48 h. Mitochondria were isolated and their protein content was assessed using a BCA assay. Equal amounts of mitochondria were lysed using 0.5% GDN at a ratio of 4.2:1 (GDN: protein by mass). A fraction of mitochondrial lysate was saved as an input control, and the rest was loaded onto GFP trap agarose beads, to pulldown proteins associated with the GFP-tagged protein of interest (trapped or imported precursor). The resulting beads conjugated to proteins associated with GFP-tagged proteins were analysed by Western blot and TMT-MS. Schematic created using BioRender. **b** Volcano plot highlighting proteins enhanced in pull-down samples from mitochondria with imported precursor (+ MTS - MTX, left) or trapped precursor (+ MTS + MTX, right). Proteins of interest are highlighted by gene names. $N = 3$ biological replicates. Statistical significance was determined using a two-tailed, unpaired $t$ test (**b**). Raw data are provided in the Supplementary Data file (**b**). BCA bicinchoninic acid, FDR false discovery rate, GDN glycol-diosgenin, LogFC $Log^2$ fold change, MTS mitochondrial targeting sequence, MTX methotrexate, NS not significant, TMT-MS tandem mass tagging mass spectrometry, TNT tunnelling nanotube.

formation can be induced by various stress factors including $H_2O_2$, serum depletion, mitochondrial stress, and cytotoxicity, which may act via the p53 pathway;[36] they alleviate cellular stress by enabling replenishment and/or removing harmful debris. Indeed, healthy mitochondria have previously been reported to transport through TNTs and rescue cells in the early stages of apoptosis[40]. Another study demonstrated that TNTs could assist in reducing cellular toxicity caused by alpha-synuclein fibrils (implicated in Parkinson's disease) in microglia, by facilitating the removal of protein aggregates and the delivery of healthy mitochondria[52]. Here, we show how direct mitochondrial import impairment leads to bidirectional mitochondrial transfer potentially via TNTs, suggestive of a widespread mechanism for the rescue of mitochondrial function. However, it is important to note that the activity of other mitochondrial transfer mechanisms, such as the release of mitochondria-containing extracellular vesicles for capture by the recipient cells cannot be excluded. Further work will be required to investigate the potential role of such mechanisms in this context.

It makes sense that the TNT response described, triggered by failing import, is established to help resolve cells incapacitated by

potentially catastrophic failure of mitochondrial biogenesis. Beyond that, we also speculate that the TOM and TIM23 complexes function as reporters of mitochondrial health; whereby, for whatever reason, reduced bioenergetic function would lead to diminished Δψ and/ or ATP, decreasing import efficiency and hence initiating rescue. In this respect, the connection between protein aggregation associated with neurodegeneration and mitochondrial import is particularly intriguing. Indeed, the intercellular transport of the Tau protein and β-amyloid, associated with Alzheimer's disease, can also occur via TNTs[53]. This could be connected to the propensity of these aggregation-prone proteins to associate with the mitochondrial import machinery. Variants of the Huntingtin protein, the causative agent of disease, and the β-amyloid precursor APP have both been shown to associate with the TOM40/TIM23 complexes[6,8,54]. Indeed, in a parallel study, we show that a Tau variant linked to neurodegeneration associates with TOM40 and brings about TNT formation[9]. Therefore, these proteins may also act by binding to and impairing the import machinery, inducing TNT formation for aggregate efflux[55–57]. On a more sinister note, perhaps this mechanism could be hijacked to spread infectious agents; whereby viruses or prions target themselves

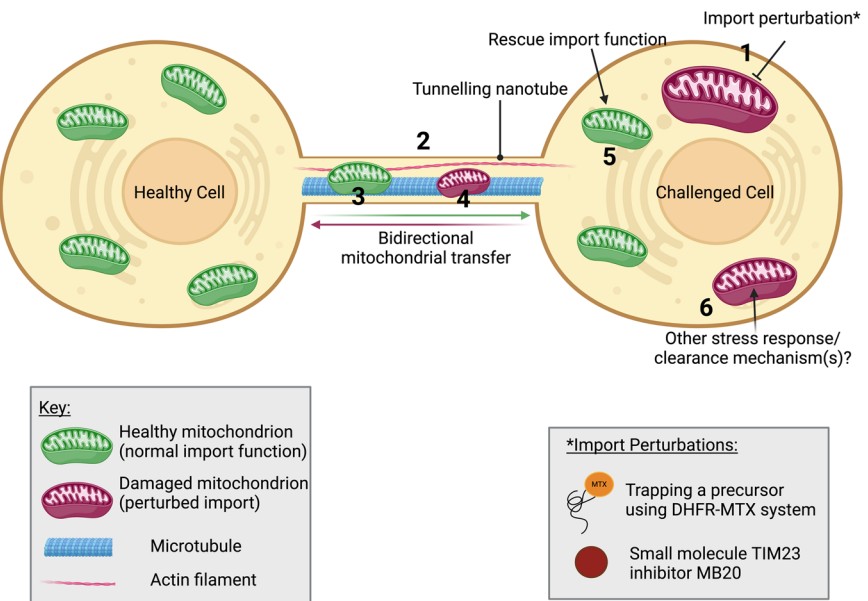

**Fig. 8 | Proposed import rescue mechanism via TNT-dependent intercellular mitochondrial transfer.** Cells subjected to import perturbations (1) induce the formation of TNTs composed of microtubules and actin filaments (2). TNTs mediate the transfer of healthy mitochondria from neighbouring cells into cells with challenged mitochondria (3), whilst also allowing clearance of damaged mitochondria to neighbouring healthy cells (4). The transfer and integration of healthy mitochondria with normal import function, and disposal of mitochondria with defective import, rescue the overall cellular mitochondrial import function (5). This rescue effect is likely enhanced by additional stress response mechanisms (6). Created using BioRender. DHFR dihydrofolate reductase, MB20 MitoBloCK-20, MTX methotrexate, TIM23 translocase of the inner mitochondrial membrane 23, TNT tunnelling nanotube.

to the mitochondrial import machinery and hitch a ride through TNTs to other cells[48,58,59].

What emerges from this study is a concept for the rescue of mitochondrial import function involving intercellular mitochondrial transport possibly by TNTs (Fig. 8). This process has only now come to light as the analysis of protein import, and import failure, has been largely restricted to the unicellular eukaryote *Saccharomyces cerevisiae*. It may well be one of many rescue mechanisms of multi-cellular organisms instigated by perturbation of the import apparatus, including the stress response to unfolded precursors in the cytosol[11,12]. It will be very interesting to see how this mechanism is manifested in vivo – to what extent and for what purpose. Mitochondrial import dysfunction underpins a wide range of devastating and often fatal diseases[10,60,61], in which cases the rescue process is presumably disabled, ineffective or overwhelmed.

While it is not exactly clear how this rescue response is instigated following import perturbation, the proteomic results present a few clues that could be addressed in future studies. A greater understanding of the formation and functions of TNTs might provide exciting opportunities for biological engineering and medical applications. For example, modulating this process genetically, or with small molecules, as we have demonstrated, could enable targeted intercellular mitochondrial transport in vivo, and approaches to mitochondrial replacement therapy for cell and tissue regeneration.

## Methods
### Reagents
All chemicals were of the highest grade of purity commercially available and purchased from Sigma, UK unless stated otherwise. Aqueous solutions were prepared in ultrapure water, while for non-aqueous solutions, ethanol or DMSO was used as solvent instead. Antibodies were all commercially available and used following suppliers' recommended protocol or validation profile for the given application unless otherwise stated. Antibodies used in Western blotting (antibody, supplier, catalogue number, dilution): DRP1, BD Biosciences, 6111113, 1:1000; phospho-DRP1 S616, CST, 4494, 1:1000; β-actin, Sigma, A2228,

1:10000; OPA1, Abcam, ab42364, 1:500; MFF, Proteintech, 17090-1-AP, 1:500; FIS1, Proteintech, 10956-1-AP, 1:1000; GFP, Sigma, G1544, 1:2000. Antibodies used in immunocytochemistry (antibody, supplier, catalogue number, dilution): Tubulin-α, BioRad, MCA78G, 1:400; β-actin, Sigma, A2228, 1:1000

### Generation of constructs
Constructs were generated by standard cloning techniques. Briefly, PCR reactions were carried out using Q5 High Fidelity Hot Start DNA Polymerase (New England Biolabs (NEB)), using 20 pmol primers and 200 pg template DNA, as per manufacturers' instructions. PCR products were purified using QIAquick PCR Purification Kit (QIAgen). Restriction digest reactions were carried out using NEB restriction enzymes at 37 °C for 45 min. Ligation reactions were carried out using T4 DNA Ligase (NEB) overnight at 16 °C. Transformation was carried out in *Escherichia coli* cells (α-select, XL1-Blue, or BL21-DE3 cells were used, depending on application; all originally sourced from NEB) for 30 min on ice, followed by heat shocking (45 s, 42 °C), and a further 15 min on ice. Cells were recovered by incubating in LB media at 37 °C for 1 h and then plated on LB-agar plates containing the appropriate antibiotic. Plasmids were prepared by mini or maxi preps using commercially available kits (Qiagen and Promega, respectively) following manufacturers' instructions, and verified by DNA sequencing using Eurofins Genomics TubeSeq service. Protein sequences are available in Supplementary Data 1.

### Protein purification
Protein expression was carried out exactly as described previously[21]. A single colony of transformed BL21 (DE3) bacteria was grown as a pre-culture in LB with appropriate antibiotics overnight (37 °C; 200 rpm). Pre-cultures were used to inoculate a secondary culture at a 1:100 dilution, in 2X YT supplemented with the appropriate antibiotics. Secondary cultures were grown until the mid-log phase, then induced with 1 mM IPTG or 0.2% (w/v) Arabinose and grown for a further 3 h. Cells were harvested by centrifugation (15 min; 6000 × g), resuspended in TK buffer (20 mM TRIS base, 50 mM KCl; pH 8.0), cracked in

a cell disruptor (Constant Systems; 2 cycles at 25 kpsi), and clarified by centrifugation (45 min; 103,500 × g).

GST tagged Recombinant Perfringolysin (rPFO) was produced as described previously[22]. The supernatant (soluble fraction) was loaded onto a 5 mL GSTrap 4B column (GE Healthcare) and the column was washed with TK buffer until the absorbance of the flow through ceased to decrease any further. The peptide was eluted using 10 μM reduced glutathione, prepared fresh. Eluted fractions were pooled and loaded onto a 5 mL anionic exchanger (Q- column; GE Healthcare). A salt gradient of 0–1 M was applied over 20 min and the protein was eluted in 5 mL fractions. Excess salt was removed by spin concentration, followed by dilution in TK buffer. The fractions containing the protein were confirmed by SDS-PAGE with Coomassie staining, then pooled and spin concentrated. The final protein concentration was determined based on an extinction coefficient of 117,120 M$^{-1}$ cm$^{-1}$. The protein was aliquoted, snap-frozen, and stored at −80 °C until required. For each assay, a fresh aliquot was thawed, used immediately, and discarded afterwards, due to the instability of this protein[62].

GST-Dark Peptide was prepared as described previously[21]. The supernatant was loaded onto a GSTrap 4B column and purification was carried out exactly as for rPFO but without the ion exchange chromatography. Analysis, yield, and freezing were carried out exactly as for rPFO. Protein concentration was determined based on an extinction coefficient of 48,360 M$^{-1}$ cm$^{-1}$.

His tagged Su9-EGFP-pep86 was prepared as described previously[22]. Inclusion bodies (insoluble fraction) were solubilised in TK buffer supplemented with 6 M urea, before loading onto a 5 mL HisTrap HP column (GE Healthcare). The protein was eluted in 300 mM imidazole. Eluted fractions containing the desired protein were pooled and loaded onto a 5 mL cationic exchanger (S- column; GE Healthcare). A salt gradient of 0–1 M was applied over 100 mL and the protein was eluted in 5 mL fractions. Excess salt was removed by spin concentration, followed by dilution in TK buffer containing 6 M urea. Analysis, yield, and freezing were carried out exactly as for rPFO. Protein concentration was determined based on an extinction coefficient of 28,880 M$^{-1}$ cm$^{-1}$.

His-SUMO--Su9-ACP1-D-ACP1-D-DHFR-Myc was produced as described previously[3]. It was expressed by induction with IPTG, and cells were grown for a further 18 h before harvesting. Subsequently, proteins from the soluble fraction were purified using a Nickel column, and finally, contaminants were removed by size exclusion chromatography using a HiLoad 16/60 Superdex gel filtration column (Cytiva, UK). The SUMO tag was cleaved from the protein using Ulp1 protease following purification, and the eluted sample was passed through a Nickel column to separate the SUMO-His tag from the protein. Protein concentration was determined based on an extinction coefficient of 42,400 M$^{-1}$ cm$^{-1}$.

## Cell culture

HEK293T (ECACC) and glucose-grown HeLa cells (HeLaGLU; ATCC) were maintained in Dulbecco's Modified Eagle's Medium (DMEM; Gibco; 41965039) supplemented with 10% (v/v) foetal bovine calf serum (FBS; Invitrogen) and 1% (v/v) penicillin-streptomycin (P/S; Invitrogen). When OXPHOS dependence was required, HeLa cells were cultured in galactose medium (HeLaGAL; ATCC) consisting of DMEM without glucose (Gibco; 11966025) supplemented with 10 mM galactose, 1 mM sodium pyruvate, 10% FBS and 1% P/S. Cells were cultured in galactose media for at least 3 weeks before experiments on HeLaGAL cells. Cells were maintained in T75 ventilated flasks in humidified incubators at 37 °C with 5% CO$_2$. Cell lines were regularly mycoplasma tested using Eurofins mycoplasma testing service.

## Cell transduction by transfection

HeLa cells were plated and grown up to ~70–80% confluency. At this point, cells were transfected with 1 μg (per 35 mm dish) of the desired DNA using Lipofectamine 3000 reagent (Thermo Fisher Scientific), at a 1:1.5 ratio of DNA: Lipofectamine, following the manufacturer's protocol. Cells were then grown for a further 24–72 h before experimental analysis.

## Cell transduction by Lentiviral infection

Lentiviral particles were produced in HEK293T cells by addition of a mixture of DNAs (27.2 μg DNA to be produced, and packaging vectors pMDG2 (6.8 μg) and pAX2 (20.4 μg)) and 1 μg/μL pEI transfection reagent in OptiMEM media (Gibco) to HEK293T cells in a T75 flask, followed by incubation for 6 h at 37 °C, 5% CO$_2$. Media was then changed to complete DMEM, and cells were incubated for 72 h to allow lentivirus particle production. Lentiviral particles were harvested at 48 and 72 h for maximum yield, pooled, spun down at 4000 × g for 5 min to remove dead cells, and concentrated by adding Lenti-X concentrator (Takara Bio) at a 1:3 ratio and incubating at 4 °C for at least 1 h. Lentivirus was then pelleted by centrifugation at 4000 × $g$ for 45 min. Pellets were resuspended in plain DMEM at 1:50 of initial supernatant volume and aliquoted and stored at −80 °C until required. For infection, concentrated lentivirus (volume optimised by titration of each fresh batch) was added dropwise to cell media when cells were at ~70–80% confluency and incubated for 24–72 h before experimental analysis.

## Precursor trapping

For precursor trapping experiments, the DHFR–MTX affinity system was utilised, as has been characterised previously[3,17,18,63]. For chronic precursor trapping, cells were pre-treated with 100 nM MTX by adding it dropwise to the cell medium the night before transduction. The next day, cells were subjected to the over-production of Su9-EGFP–DHFR, Su9-mScarlet-DHFR, or Su9-eqFP670-DHFR either by transfection or infection (as detailed in the main text) and cells were incubated for the time detailed in figure legends prior to experimental analysis. Cells expressing Su9-EGFP/mScarlet/eqFP670–DHFR without MTX (imported precursor, no trap) were used as a control for the effects of trapping, and cells expressing EGFP/mScarlet/eqFP670–DHFR (no MTS) with or without MTX were used to control for non-trapping related effects of MTX. For acute trapping, cells were permeabilised using 3 nM rPFO prior to incubation (RT, 10 min) with 1 μM trap substrate (Su9-ACP1-D-ACP1-D-DHFR-Myc), in the presence of 100 nM MTX, prior to experimental analysis. Cells incubated with 1 μM Su9-ACP1-D-ACP1-D-DHFR-Myc in the absence of MTX (imported precursor, no trap) were used as a control for the effects of acute trapping.

## Total protein cell lysis

For extraction of total protein lysate, cells were washed extensively with HBSS, and RIPA buffer (Sigma, supplemented with 1 mM PMSF) was added (200 μL for ~35 mm surface, scaled up or down appropriately). Cells were scraped on ice into Eppendorf tubes, which were incubated for 1 h on a rotating wheel at 4 °C, before spinning down at 10,000 × g for 15 min at 4 °C. The supernatant (containing proteins) was stored at −20 °C. If protein concentration was required, this was obtained by carrying out a BCA assay (Pierce$^{TM}$ BCA Protein Assay Kit, Thermo Fisher Scientific), following the manufacturer's instructions and using BSA as a standard.

## Western blotting

Following total protein extraction, protein samples were heated to 95 °C for 5 min in the presence of LDS supplemented with 50 mM DTT. 30 μg of total protein was loaded on 4–12% BOLT gels (Thermo Fisher Scientific), separated (200 V, 24 min), and transferred onto polyvinylidene difluoride (PVDF) membranes (activated with methanol; Thermo Fisher Scientific) with transfer buffer (336 mM tris, 260 mM glycine, 140 mM tricine, 2.5 mM EDTA) using a semi-dry Pierce Power Station transfer system (Thermo Fisher Scientific; 25 V, 2.5 mAmp,

10 min). Membranes were blocked for 1 h in milk or BSA (5% w/v in TBS-T: 20 mM TRIS, 1.5 M NaCl, 0.1% (v/v) Tween-20 (pH 7.6)) and incubated in 5% milk or BSA (for phospho- antibodies) containing the appropriate primary antibody (see figure legends; 4 °C, overnight). Membranes were washed extensively with TBS-T and probed with the appropriate secondary antibody in 2.5% milk or BSA (RT, 1 h). Membranes were washed with TBS-T, incubated with ECL substrate (GE Healthcare), and developed using an Odyssey Fc Imaging System (LI-COR). Analysis and quantification were carried out using Image Studio Lite software. For transparency, full, uncropped Western blot images are shown in the Source Data file.

### Mitochondrial isolation
Confluent cells were harvested by trypsinisation, pelleted, and washed extensively with HBSS. Pellets were frozen overnight at −80 °C and thawed the next day, to weaken membranes. Subsequently, mitochondrial isolation was performed using Mitochondrial Isolation Kit for Cultured Cells (Abcam; ab110170) following the manufacturer's instructions. Mitochondrial protein concentration was calculated by BCA assays using BSA as a standard.

### Immunoprecipitation
Following mitochondrial isolation, mitochondria were gently lysed using 4.2 g GDN: 1 g protein in IP buffer (0.1 M TRIS-HCl, 0.15 M NaCl, phospholipids (0.03 mg/ml PE; 0.03 mg/ml PG; 0.09 mg/ml PC), 1X cOmplete ULTRA Protease Inhibitor Cocktail). GFP-tagged proteins were isolated using 10 μL GFP trap beads (Chromotek). The supernatant (lysed mitochondrial sample) was incubated on a rotating wheel with beads overnight at 4 °C. Subsequently, beads were washed in IP buffer. After washing, the supernatant was removed, and samples were analysed by Western blotting or mass spectrometry.

### Proteomic analysis
**TMT labelling and high pH reversed-phase chromatography.** Immuno-isolated samples were reduced (10 mM TCEP, 55 °C for 1 h), alkylated (18.75 mM iodoacetamide, RT for 30 min) and then digested from the beads with trypsin (2.5 μg trypsin; 37 °C, overnight). The resulting peptides were then labelled with TMTpro sixteen-plex reagents according to the manufacturer's protocol (Thermo Fisher Scientific) and the labelled samples were pooled and desalted using a SepPak cartridge, according to the manufacturer's instructions (Waters). Eluate from the SepPak cartridge was evaporated to dryness and resuspended in buffer A (20 mM ammonium hydroxide, pH 10) before fractionation by high pH reversed-phase chromatography using an Ultimate 3000 liquid chromatography system (Thermo Fisher Scientific). In brief, the sample was loaded onto an XBridge BEH C18 Column (130 Å, 3.5 μm, 2.1 mm × 150 mm, Waters) in buffer A and peptides were eluted with an increasing gradient of buffer B (20 mM Ammonium Hydroxide in acetonitrile, pH 10) from 0 to 95% over 60 min. The resulting fractions (6 in total) were evaporated to dryness and resuspended in 1% formic acid before analysis by nano-LC MSMS using an Orbitrap Fusion Lumos mass spectrometer (Thermo Fisher Scientific).

**Nano-LC mass spectrometry.** High pH RP fractions were further fractionated using an Ultimate 3000 nano-LC system in line with an Orbitrap Fusion Lumos mass spectrometer (Thermo Fisher Scientific). In brief, peptides in 1% (vol/vol) formic acid were injected into an Acclaim PepMap C18 nano-trap column (Thermo Fisher Scientific). After washing with 0.5% (v/v) acetonitrile 0.1% (v/v) formic acid, peptides were resolved on a 250 mm × 75 μm Acclaim PepMap C18 reverse-phase analytical column (Thermo Fisher Scientific) over a 150 min organic gradient, using 7 gradient segments (1–6% solvent B over 1 min, 6–15% B over 58 min, 15–32% B over 58 min, 32–40% B over 5 min, 40–90% B over 1 min, held at 90% B for 6 min and then reduced to 1% B over 1 min) with a flow rate of 300 nl min⁻¹. Solvent A was 0.1% formic acid and Solvent B was aqueous 80% acetonitrile in 0.1% formic acid. Peptides were ionised by nano-electrospray ionisation at 2.0 kV using a stainless-steel emitter with an internal diameter of 30 μm (Thermo Fisher Scientific) and a capillary temperature of 300 °C.

All spectra were acquired using an Orbitrap Fusion Lumos mass spectrometer controlled by Xcalibur 3.0 software (Thermo Fisher Scientific) and operated in data-dependent acquisition mode using an SPS-MS3 workflow. FTMS1 spectra were collected at a resolution of 120,000, with an automatic gain control (AGC) target of 200,000 and a max injection time of 50 ms. Precursors were filtered with an intensity threshold of 5000, according to charge state (to include charge states 2–7) and with monoisotopic peak determination set to Peptide. Previously interrogated precursors were excluded using a dynamic window (60 s +/-10 ppm). The MS2 precursors were isolated with a quadrupole isolation window of 0.7 m/z. ITMS2 spectra were collected with an AGC target of 1000, max injection time of 70 ms and CID collision energy of 35%.

For FTMS3 analysis, the Orbitrap was operated at 50,000 resolution with an AGC target of 50,000 and a max injection time of 105 ms. Precursors were fragmented by high energy collision dissociation at a normalised collision energy of 60% to ensure maximal TMT reporter ion yield. Synchronous Precursor Selection (SPS) was enabled to include up to 10 MS2 fragment ions in the FTMS3 scan.

**Data analysis.** The raw data files were processed and quantified using Proteome Discoverer software v2.4 (Thermo Fisher Scientific) and searched against the UniProt Human database (downloaded January 2021: 169297 entries) using the SEQUEST HT algorithm. Peptide precursor mass tolerance was set at 10 ppm, and MS/MS tolerance was set at 0.6 Da. Search criteria included oxidation of methionine (+ 15.995 Da), acetylation of the protein N-terminus (+ 42.011 Da) and Methionine loss plus acetylation of the protein N-terminus (−89.03 Da) as variable modifications and carbamidomethylation of cysteine (+ 57.0214) and the addition of the TMTpro mass tag (+ 304.207) to peptide N-termini and lysine as fixed modifications. Searches were performed with full tryptic digestion and a maximum of 2 missed cleavages were allowed. The reverse database search option was enabled and all data was filtered to satisfy a false discovery rate of 5%.

### Light microscopy
**Fixed cell confocal microscopy.** For staining with dyes, cells on glass coverslips were incubated with 100 nM MitoTrackerRed CMXRos (Invitrogen) for 30 min at 37 °C then washed twice in HBSS. Cells were then incubated with 5 μg/mL Wheat Germ Agglutin (WGA) Alexa Fluor 633 (Thermo Fisher Scientific) for 10 min at RT then washed three times in HBSS followed by fixation (protocol adapted from[64]). Coverslips were washed three times in HBSS and then incubated in fixative 1 (2% (w/v) paraformaldehyde (PFA), 0.05% (w/v) glutaraldehyde, 0.2 M HEPES in PBS) for 20 min at RT. Fixative 1 was removed and coverslips were incubated with fixative 2 (4% (w/v) PFA, 0.2 M HEPES in PBS) for a further 20 min at RT. Coverslips were washed once with 100 mM glycine in PBS for 4 min followed by three quick washes in PBS. Coverslips were dipped in ddH₂O before mounting using Fluoromount-G mounting medium (Thermo Fisher Scientific). Cells were imaged using a Leica confocal microscope (SP8) with 'lightning' adaptive image restoration, and the Leica Application Suite X (LAS X) software platform. Laser lines used were 488, 562, and 633 nm, with the gain set to allow maximum sensitivity without saturation, and z-stacks were taken.

For immunocytochemistry, following fixation and quenching (as above), cells were permeabilised by incubation in 0.1% (v/v) Triton-X in PBS for 5 min, washed in PBS and blocked with 3% (w/v) BSA in PBS for 30 min. Coverslips were incubated on a drop of primary antibody in 3% BSA overnight at 4 °C. Coverslips were washed extensively in PBS and

incubated with the appropriate secondary antibody (2 h, RT) in 3% BSA, then washed again. Coverslips were dipped in ddH2O and mounted using Fluoromount-G. Coverslips were left to dry overnight and imaged using a Leica confocal microscope (SP5II or SP8) and the LAS X software platform. Laser lines used were 405, 488, 562, and 633 nm, with gain set to allow maximum sensitivity without saturation, and z-stacks were taken.

**Live cell confocal microscopy.** Cells were seeded onto 35 mm glass-bottomed dishes (Corning). Immediately before imaging, cells were washed in HBSS, transferred to imaging media (10% FBS, 2 mM L-Glutamine, 10 mM D-Galactose, 1 mM Sodium Pyruvate, 40 mM HEPES in phenol red-free DMEM), and mitochondria were stained with 25 nM TMRM for 30 min at 37 °C before imaging (TMRM was retained in buffer throughout imaging). Cells were imaged immediately using a Leica SP8 confocal microscope and the LAS X software platform at 37 °C, using 488 and 562 nm laser lines.

**Live cell spinning disk microscopy.** Cells were seeded onto 35 mm glass-bottomed dishes (Corning). Immediately before imaging, cells were incubated with MitoTracker Green for 30 min at 37 °C, washed three times in HBSS and transferred into imaging media (as above). Imaging was performed using an Olympus IXplore SpinSR system at 37 °C, using 488 and 561 nm laser lines. A z-stack was taken with 10 z-slices, taken every 5 min with 10 time points.

**Live cell structured illumination microscopy.** Transfected cells were grown to confluency on glass coverslips before staining with 100 nM MitoTracker Green for 30 min at 37 °C in a complete growth medium. Coverslips were washed in HBSS and incubated in a complete growth medium in Attofluor Cell Chambers (Invitrogen) for live imaging using super-resolution 3D structured illumination microscopy (SR-3D SIM) on an OMX v4 microscope (GE Healthcare) at 37 °C, using 488 and 568 nm laser lines and a 60x NA PLAPON oil objective. Image stacks were acquired with an optical spacing of 125 nm. Super-resolution images were reconstructed and registered with softwoRx 7.0 (GE Healthcare). The necessary optical transfer functions and channel registrations were acquired shortly before.

**Light microscopy image analysis.** All image analysis was performed using the FiJi image processing package[65,66]. Macros were written by Dr Stephen Cross (Wolfson Bioimaging Facility, University of Bristol), within the FiJi plugin Modular Image Analysis (MIA) package version 0.21.0, which is publicly accessible on GitHub with a linked version-specific DOI from Zenodo[67,68]. Where data was analysed manually, data was blinded before manual analysis.

Mitochondrial pre-processing, as well as branch and network analysis, was carried out as described previously[69,70], using a FiJi plugin adapted from the mitochondrial network analysis (MiNA) toolset[71]. Briefly, for pre-processing, cells of interest were outlined, and the outside cleared, before z-stack max intensity projection, contrast enhancement and background subtraction. Median, unsharp, and tubeness filters were applied. For branch/network analysis, images were binarised and skeletonised, and the skeleton was analysed to obtain information on the branches and pixels, allowing for analysis of mitochondrial branching as a readout of mitochondrial network complexity and fragmentation.

Analysis of mitochondrial width was carried out by manual measurement. Data was blinded and the same region of mitochondria was measured in each cell (100 × 100 pixels at 2 o'clock relative to the nucleus) to maintain unbiased results. Five measurements were taken across each mitochondrion (parallel to cristae) and averaged to obtain an average width for the given mitochondrion. An equal number of mitochondria were analysed for each condition. Circularity analysis was carried out using a macro that classifies mitochondria based on

their shape, providing a circularity index, where a value of 1 represents a perfect circle.

Association analysis was carried out using a macro, on images acquired by SIM. Mitochondria (channel 1) are detected in 2D within a region of interest (ROI, i.e., a transfected cell) and the pixel intensity of the trapping substrate (channel 2) is measured as a 4-pixel wide strip at the edge of the mitochondrion. The background channel 2 intensity (intensity at 10−14 pixels from the edge of the mitochondrion) is subtracted from the raw intensity surrounding the mitochondrion, to account for differing expression levels between cells. This provides values for the intensity directly surrounding mitochondria, and after thresholding, these values highlight the proportion of mitochondria within a cell or population of cells that have trapped protein on their outside.

Mitochondrial mixing was analysed on FiJi using the MIA package. For each cell, a global threshold of the sum of green and red channels was calculated. The thresholded image was then used as a mask for the colocalisation measurement of green and red channels. An unpaired *t* Test was performed on collected Pearson's Correlation Coefficient (PCC) values (where 0 represents no correlation (no mixing) and 1 represents perfect positive correlation (mixing of mitochondrial markers)).

### MitoLuc assay

MitoLuc assays were carried out exactly as described previously[22]. Briefly, cells over-producing Cox8a−11S (and, where applicable, with other relevant proteins/treatments, see figure legends for details) and plated on standard white flat-bottom 96-well plates were washed with HBSS and incubated in HBSS imaging buffer (HBSS supplemented with 5 mM D-(+)-glucose,10 mM HEPES (Santa Cruz Biotechnology, Germany), 1 mM MgCl$_2$, 1 mM CaCl$_2$; pH 7.4). A fluorescence read was taken at 605/670 nm using monochromators with the gain set to allow maximum sensitivity without saturation, using a BioTek Synergy Neo2 plate reader (Agilent, UK). Cells were transferred to MitoLuc assay master mix (225 mM mannitol, 10 mM HEPES, 2.5 mM MgCl$_2$, 40 mM KCl, 2.5 mM KH$_2$PO$_4$, 0.5 mM EGTA; pH 7.4) supplemented with 5 mM succinate, 1 μM rotenone, 0.1 mg/mL creatine kinase, 5 mM creatine phosphate, 1 mM ATP, 0.1% (v/v) Prionex, 3 nM rPFO (purified in house), 20 μM GST-Dark (purified in house), and 1:800 furimazine (Nano-Glo® Luciferase Assay System; Promega). A baseline read of 30 s of background luminescent signal was taken before injection of a purified substrate precursor protein (Su9-EGFP-pep86) to 1 μM final concentration, followed by a further bioluminescence read corresponding to import, lasting 30 min (or until saturation). Bioluminescence was read using a BioTek Synergy Neo2 plate reader or a CLARIOstar Plus plate reader (BMG LabTech, UK) without emission filters with the gain set to allow maximum sensitivity without saturation, and with acquisition time of 0.1 s per well. Row mode was used, and reads were taken every 6 s or less, with wells in triplicate. For analysis, traces were normalised to eqFP670 fluorescence to control for variations in mitochondrial Cox8a-11S. Then, technical replicates were averaged, average values were normalised to the maximum amplitude of the run, and finally, data were normalised to the ± MTX (- MTS) control (where relevant).

### TMRM membrane potential analysis

Analysis of mitochondrial membrane potential was done based on a published protocol[72]. Briefly, transfected cells were grown to confluency on 35 mm glass-bottomed dishes (Corning). Cells were washed in HBSS and incubated in HBSS imaging buffer with 25 nM TMRM for 30 min at 37 °C. Cells were imaged immediately, with TMRM retained in the buffer, using a Leica SP8 confocal microscope and the LAS X software platform, at 37 °C. Images were captured over 2 min, and then 10 μM carbonyl cyanide m-chlorophenylhydrazone (CCCP) was added to dissipate the ΔΨ and control for background. For analysis, an

average of the first 10 frames was taken, and then the mitochondria were thresholded and their intensity measured. The average of the final three frames (after CCCP treatment) was subtracted as a control and the intensity of the subtracted image was re-measured.

## Mitochondrial Stress Test (Seahorse OCR Assay)

Cells were seeded in 6-well plates (200,000 cells/well) and transfected appropriately, as described in figure legends. The day before the assay, cells were detached by trypsinisation, counted, and seeded at 10,000 cells/well in 96-well Seahorse XF cell culture plates (Agilent). The sensor cartridges were hydrated overnight with tissue culture grade $H_2O$ in a non-CO2 incubator at 37 °C as per the manufacturer's instructions. On the day of the assay, $H_2O$ in the sensor plate was replaced with Seahorse XF Calibrant (Agilent) and cells were washed with HBSS and incubated in Seahorse media (Seahorse XF assay medium (Agilent), 1 mM pyruvate (Agilent), 2 mM glutamine (Agilent), 10 mM D-(+)-galactose). Both sensor and cell plates were incubated in a non-$CO_2$ incubator at 37 °C for 1 h before the assay. The sensor plate was loaded with oligomycin (15 μM for 1.5 μM final concentration in wells; injector A), CCCP (5 μM for 0.5 μM final; injector B), and antimycin A and rotenone (5 μM/5 μM for 0.5 μM/0.5 μM final; injector C). The sensor plate was calibrated before loading cells and running a mitochondrial stress test using a Seahorse XFe96 (Agilent). Following assays, cells were washed and fixed in 1% (v/v) acetic acid in methanol at −20 °C overnight for SRB assays, which were used for normalising data to protein content.

## Sulforhodamine B (SRB) Assay

SRB assays were used to determine cell mass for the normalisation of seahorse data. Cells were fixed with ice-cold 1% acetic acid in methanol overnight at −20 °C. Fixative was aspirated and plates were allowed to dry at 37 °C. SRB (0.5% (w/v) in 1% (v/v) acetic acid in d$H_2O$) was added to cover cells, and plates were incubated for 30 min at 37 °C. SRB was then aspirated, and any unbound stain was removed by washing the plates four times with 1% acetic acid in d$H_2O$, before drying plates at 37 °C. The bound protein stain was solubilised by shaking incubation with 10 mM TRIS (pH 10; 15 min, RT). Absorbance was read on a microplate reader with a 544/15 nm filter.

## Co-cultures

For co-cultures, one batch of HeLaGAL cells was subjected to Cox8a-DsRed over-production (healthy mitochondria) by lentiviral infection. A second batch of cells was subjected to Su9-EGFP over-production (challenged mitochondria) by lentiviral infection and sequential precursor trapping (Su9-eqFP670-DHFR + 100 nM MTX) by transfection. After 48 h, cells were washed, detached, counted, and mixed at equal ratios, before reseeding the mixed population on coverslips. Cells were co-cultured for a further 8 h before fixation and visualisation by confocal microscopy.

## Statistical analysis

The sample size was determined based on previous research. Sample size (N) for all experiments was 3–6 biological replicates, with technical replicates or individual objects (cells/mitochondria) analysed for each biological replicate, as appropriate. Statistics were always performed on biological replicates. Normality of distribution was first tested, and statistical significance between groups was determined using unpaired Student's $t$ Test or one- or two-way ANOVA with interaction if more than two groups were analysed. Nested ANOVA was used if multiple objects were analysed per biological replicate. Following ANOVA, $p$ values were adjusted for multiple comparisons through Tukey's or Šidák's post hoc tests and differences were considered significant at a 5% level. Statistical analyses were performed using Graph Pad Prism version 9 (Graph-Pad Software, Inc., San Diego, CA, USA). Error bars on graphs always represent the standard deviation of the mean (mean ± SD). Asterisks represent significance as follows: ns = $p > 0.05$, * = $p ≤ 0.05$, ** = $p ≤ 0.01$, *** = $p ≤ 0.001$, ****$p ≤ 0.0001$. Individual points (for each biological replicate or object where appropriate, as detailed in the figure legend) are graphed to show the true spread of the data.

## Reporting summary

Further information on research design is available in the Nature Portfolio Reporting Summary linked to this article.

## Data availability

All data are available in the main text or the supplementary materials. Proteomics data are available via ProteomeXchange with the identifier PXD040098 and are included with this paper in the Supplementary Data file. MitoCOP analysis data is included in the Supplementary Data file. Protein sequences are available in the Supplementary Data file. All source data, including uncropped Western blots, are provided with this paper and are available in the Source Data file. Source data are provided with this paper.

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

## Acknowledgements

We gratefully acknowledge the access and support of the Wolfson Bioimaging Facility for imaging support, as well as Drs Stephen Cross and Richard Seager for image analysis support. We also wish to thank the University of Bristol's Proteomics Facility, specifically Dr Kate Heesom for support with TMT-MS, and Dr Phil Lewis for help with the bioinformatics analysis. This work was funded by the Wellcome Trust: through the Wellcome Trust *Dynamic Molecular Cell Biology* PhD programme (083474, HIN; 218510, EG) and a Wellcome Investigator award (104632, IC and GCP). The funders had no role in study design, data collection and interpretation, or the decision to submit the work for publication. For Open Access, the authors have applied a CC BY public copyright license to any Author Accepted Manuscript version arising from this submission. Biotechnology and Biological Sciences Research Council (BBSRC) Alert 19 equipment grant (BB/T017597/1).

## Author contributions

HIN, EG, GCP, MPD, JMH and IC designed experiments; HIN, EG, AW and WH conducted experiments; HIN, GCP, JMH and IC wrote the manuscript; IC and JMH secured funding and led the project.

## Competing interests

The authors declare no competing interests.
