## [Peer Review File · Nature Communications]

Rescue of mitochondrial import failure by intercellular organellar transferREVIEWER COMMENTS

Reviewer #1 (Remarks to the Author):

Despite the extensive studies on several stress responsive pathways against mitochondrial import stress in yeast, such mechanisms are largely uncharacterized in mammalian cells. In this manuscript, by establishing an artificial clogger-induced mitochondrial import dysfunction model in mammalian cells, the authors showed that the intercellular exchange of healthy mitochondria and dysfunctional mitochondria through tunneling nanotubes (TNTs) maintains the optimal mitochondrial import capacity against the mitochondrial import stress. Their findings are very intriguing, but following points should be considered. 1) Because this manuscript firstly describes the artificial clogger-induced mitochondrial import dysfunction model in mammalian cells, more careful validation of this assay should be conducted. 2) Although the authors performed an IP-MASS spec analysis of the mitochondrial import clogger (Fig. 4D), the validation of the hit interactors are not conducted. Therefore, mechanistic insights of TNTs formation in this context remained to be speculative. Please see following specific points.

Major points

1. In Fig. S1, the authors showed that mitochondrial targeting of the mature form of Su9-EGFP-DHFR is diminished after MTX addition-induced DHFR folding. Where is the premature (MTS-unprocessed) form? If the MTX addition prevents the import of Su9-EGFP-DHFR, the premature form should appear on the immunoblotting. Do these premature forms reside in TOM-TIM import complex in mitochondria fraction as clogged precursors?

2. In Fig. 1B, from the SIM image, the authors estimate that some mitochondria (43%) are enveloped by aggregated precursor proteins. However, this is not clearly demonstrated. It seems majority of the precursor proteins accumulates in the cytosol. In this artificial clogger assay, how much percentage of TOM-TIM complex are actually occupied with aggregated precursor proteins? The aggregated precursor proteins that coated mitochondria in Fig. 1B can be marked with ubiquitin?

3. In Fig. 1C and 1D, the authors showed that the induction of mitochondrial import cloggers results in the massive mitochondrial fragmentation through Drp1 activation. However, as shown in Fig. S7, phosphorylation of Drp1 seems to be also enhanced by MTX treatment in the absence of clogger. Are there any specific reasons that the authors only focus on Drp1? For example, OPA1 processing is not accompanied with this mitochondrial fragmentation? TMRM or MitoTracker staining seems to be preserved after the clogger induction, so the authors think that these fragmented mitochondria still maintain the mitochondrial membrane potential?

4. In Fig. 1E, the logic derived from these data (the chronic trapping vs. the acute trapping) is unclear, because the authors used a completely different method to induce import block in the acute trapping. If a rescue mechanism is expected to be operating under the chronic trapping, mitochondria should experience import problem at earlier time point also in this method. Does the short treatment of MTX (the same method utilized in the chronic trapping) block the import in this mitochondrial import monitoring assay?

5. In Fig. 3B, as the evidence of the intracellular mitochondrial transfer, the authors showed one representative cell image that contains cloggers from challenged cells and mitochondrial marker proteins from healthy cells. This is an important data to show that the intracellular mitochondrial transfer happens in this context. Therefore, the quantification is necessary. Also, in these cells, mitochondrial morphology is rescued?

6. Like nocodazole, does the ablation of any factor listed as potential interactors of clogger in Fig. 4D disrupt the TNTs formation in this context?

7. TNTs formation by mitochondrial clogger seems to be only observed in HeLa cells that are cultured in galactose medium. It might be better to include potential explanations on this point.

Reviewer #2 (Remarks to the Author):

Hi

Thank you by exploring a novel and great area of research, cell to cell communication and amplification of disease. The consequences of this manuscript can provide explanations for multiple diseases that spread cell to cell.

The approach is smart and well done to support the conclusions

Some comments for the reader include

1. The inclusion of quantification of the data specially colocalization and efficiency of the transfer
2. TNT formation time course

3. High magnification of the pictures (insets) can provide a better demonstrations of the points described
4. How Fig. S3, change respect to time?
5. Fig. S4, can be only indicated in the text.
6. Fig. 1C and S5 pictures are the same. Please provide new ones
7. Can the author provide better pictures (in my copy looks low and high intensity) for S8 and S9.
8. Is Fig. 1E the same than S12?. Provide a new one. Repeating the same figure is not OK
9. Volcano plot in S14 do not have labeled targets

But overall, this is a great manuscript

POINT BY POINT RESPONSE TO THE REVIEWER COMMENTS:

Reviewer #1 (Remarks to the Author):

Despite the extensive studies on several stress responsive pathways against mitochondrial import stress in yeast, such mechanisms are largely uncharacterized in mammalian cells. In this manuscript, by establishing an artificial clogger-induced mitochondrial import dysfunction model in mammalian cells, the authors showed that the intercellular exchange of healthy mitochondria and dysfunctional mitochondria through tunneling nanotubes (TNTs) maintains the optimal mitochondrial import capacity against the mitochondrial import stress. Their findings are very intriguing, but following points should be considered.

1) Because this manuscript firstly describes the artificial clogger-induced mitochondrial import dysfunction model in mammalian cells, more careful validation of this assay should be conducted.

We have adapted and extensively characterised the MitoLuc assay to monitor mitochondrial protein import in mammalian (HeLa) cells, and this work has now been published in the *Journal of Molecular Biology*¹. We have referred to this in the text.

2) Although the authors performed an IP-MS/MS analysis of the mitochondrial import clogger (Fig. 4D), the validation of the hit interactors are not conducted. Therefore, mechanistic insights of TNTs formation in this context remained to be speculative. Please see following specific points.

We feel that a validation of the TMT proteomic data is beyond the scope of this paper, but we include this information for speculation purposes. We make this point clear in the discussion. We could remove this data; however, its inclusion could prove useful to fellow researchers, encouraging further interest and follow-up studies about what factors could be involved in the rescue process. Therefore, we suggest that (assuming the data passes a technical review for quality control) it be included in the revised version. See also note below* in response to point 6.

Major points

1. In Fig. S1, the authors showed that mitochondrial targeting of the mature form of Su9-EGFP-DHFR is diminished after MTX addition-induced DHFR folding. Where is the premature (MTS-unprocessed) form? If the MTX addition prevents the import of Su9-EGFP-DHFR, the premature form should appear on the immunoblotting. Do these premature forms reside in TOM-TIM import complex in mitochondria fraction as clogged precursors?

Unfortunately, the mature protein (-MTS) migrates at the same apparent MW as the full-length precursor (+MTS) (Fig. S1; both versions are visible in the cytosolic fraction). This is quite common for mitochondrial and bacterial precursor proteins, especially when the proteins are quite long as in this case (when the mature and precursor proteins' MWs are too close to distinguish).

Therefore, we cannot tell if the import machinery is clogged by the mature or precursor protein. If the trapped substrate is long enough then the MTS can reach the matrix and be processed (as we have previously shown²; Fig. 1A). So, the TOM-TIM is probably clogged by a mixture of mature and precursor protein, depending on the efficiency of MTS progression when the precursor is trapped. This point has been elaborated in the text.

2. In Fig. 1B, from the SIM image, the authors estimate that some mitochondria (43%) are enveloped by aggregated precursor proteins. However, this is not clearly demonstrated.

Thanks for pointing this out. We have now expanded Fig. 1B to more clearly demonstrate the association of the precursor with some mitochondria and the lack of association with others. We have also described (in the figure legend as well as in more detail in the methods) how the image analysis was carried out towards the conclusion that 43% of mitochondria are surrounded by precursor; this is also highlighted in Fig. S3 and S4.

It seems majority of the precursor proteins accumulates in the cytosol. In this artificial clogger assay, how much percentage of TOM-TIM complex are actually occupied with aggregated precursor proteins? The aggregated precursor proteins that coated mitochondria in Fig. 1B can be marked with ubiquitin?

As the reviewer mentions, we can see from Western blotting, confocal imaging, and SIM experiments (Fig. S1, S2, 1B, S3 and S4) that in the presence of Su9-EGFP-DHFR +MTX, some of the trapping protein is accumulated at mitochondria, with the remainder in the cytosol. Our investigation of the impact of trapping on import by MitoLuc assays in the presence of Nocodazole (Fig. 4C and S17) highlight a reduction in import of ~33%. When cells were subjected to acute trapping for 10 min (by permeabilisation and exogenous addition of the trapping precursor +MTX; Fig. 1F and S7) we saw a reduction of ~37%. When cells were exposed to trapping for 2 h (by transfection with DNA encoding the trap protein and incubation with MTX for 2 h), there is a reduction in import of ~48% (Fig. 1G and S8). These data are in line with the quantification from SIM, which indicated that ~43% of mitochondria were surrounded by associated precursors (Fig. 1B, S3, and S4). From this, we can conclude that the trap blocks between one-third and one-half of the cell's TOM-TIM complexes. This has now been discussed in the text.

3. In Fig. 1C and 1D, the authors showed that the induction of mitochondrial import cloggers results in the massive mitochondrial fragmentation through Drp1 activation. However, as shown in Fig. S7, phosphorylation of Drp1 seems to be also enhanced by MTX treatment in the absence of clogger.

We agree with the reviewer that phosphorylation of DRP1 at S616 is also enhanced in our control condition (-MTS +MTX) and have reconsidered our quantification strategy for these Western blots to ensure we account for the off-target effects of MTX treatment.

We decided that the best strategy to account for differences in p-DRP1 that are the result of off-target effects attributable to MTX treatment (and not to import defects) would be to work out a ratio of p-DRP1 S616/Total DRP1, where each of the +MTX conditions is normalised to their respective -MTX counterparts. We have also increased the N (biological replicates) for this experiment to further substantiate the results.

The results of this analysis show that, although there is a slight increase in p-DRP1 S616/total DRP1 in the -MTS +MTX condition, this is not significant. However, upon trapping the precursor with the +MTS +MTX condition, there is a significant increase in the phosphorylation of DRP1 at S616. We have now detailed this in the text, and the new quantification is shown in Fig 1E. In addition, we have added the Western blot to the main figure alongside the quantification (Fig 1D).

Are there any specific reasons that the authors only focus on Drp1? For example, OPA1 processing is not accompanied with this mitochondrial fragmentation?

We initially chose to investigate DRP1 as it is a very well-studied and well-characterised fission protein. And, furthermore, phosphorylation at Serine 616 has been well-characterised as a fission-promoting post-translational modification.

However, we agree with the reviewer's comment that it would be interesting to investigate changes in other fission-related proteins in response to precursor trapping. We, therefore, carried out further experiments to investigate the abundance and processing of OPA1, as well as the abundance of MFF and FIS1. We saw no changes in any of these proteins that were not attributable to the off-target effects of MTX treatment, indicating that the increased mitochondrial fragmentation we observed by SIM can be attributed to increased DRP1 phosphorylation at S616. These data are described in the text and shown in Fig. S6.

TMRM or MitoTracker staining seems to be preserved after the clogger induction, so the authors think that these fragmented mitochondria still maintain the mitochondrial membrane potential?

We carried out additional experimentation to investigate the impact of trapping on the membrane potential and respiratory function of the cells. Analysis of TMRM intensity by confocal imaging showed no change in the membrane potential of cells exposed to chronic trapping, and mitochondrial stress tests using a Seahorse respirometer showed that there was also no change

in the cells' respiratory function. This is likely due to the TNT-dependent rescue mechanism rescuing the mitochondrial dysfunction by intercellular mitochondrial transfer. The data is now presented in Fig. S9 and has been described in the text.

4. In Fig. 1E, the logic derived from these data (the chronic trapping vs. the acute trapping) is unclear, because the authors used a completely different method to induce import block in the acute trapping. If a rescue mechanism is expected to be operating under the chronic trapping, mitochondria should experience import problem at earlier time point also in this method. Does the short treatment of MTX (the same method utilized in the chronic trapping) block the import in this mitochondrial import monitoring assay?

Good point. We carried out MitoLuc import assays on cells overproducing the trapping precursor with a time course of MTX treatment. We treated the cells without MTX (no trap) or with MTX for 48 h (chronic trap), 4h, 2h, 1h, and 10 min (trapping time course). After 10 min and 1 h of MTX exposure, we did not see any response (presumably this is too short a time to have any effect with trapping in this manner as it relies on the cells producing enough new trap protein in the given timeframe to block import sites). However, when we trapped for 2 h, we saw a significant reduction in import amplitude of ~48%. Then, after 4 h of trapping, the import amplitude was comparable to the no trap control, suggesting that in the latter 2 hours of trapping, the rescue response rescued the cells' import function. This indicates that within a 2-hour period, the cells can sense the import defect and respond to it by forming TNTs and transferring mitochondria. This data is shown in Fig. 1G and S8.

The imaging data also highlights that as well as being fast (see above) the rescue response is also continuous. In addition to the rescue being activated within the first 4 h of exposure to the trap (Fig. 1G and S8), we observe TNTs connecting cells at 48 hours (Fig 2C, 3A, C, D, and E, and movie S1), suggesting that the rescue response is continually happening. This point has now been clarified in the text.

5. In Fig. 3B, as the evidence of the intracellular mitochondrial transfer, the authors showed one representative cell image that contains cloggers from challenged cells and mitochondrial marker proteins from healthy cells. This is an important data to show that the intracellular mitochondrial transfer happens in this context. Therefore, the quantification is necessary. Also, in these cells, mitochondrial morphology is rescued?

We performed correlation analysis to quantify the degree of co-localisation between mitochondrial markers in cells with mitochondrial mixing compared to controls. The data confirmed that, following mitochondrial mixing, mitochondrial markers were co-localised, indicating that mitochondria transferred via TNTs successfully integrated into the mitochondrial network of the recipient cell. This is now described in the text (and methods) and shown in Fig. S12.

6. Like nocodazole, does the ablation of any factor listed as potential interactors of clogger in Fig. 4D disrupt the TNTs formation in this context?

We identify, through proteomics, other factors that could be involved in TNT formation, and present them as potential leads for further analysis. See also note above* (point 2).

7. TNTs formation by mitochondrial clogger seems to be only observed in HeLa cells that are cultured in galactose medium. It might be better to include potential explanations on this point. When HeLa cells are cultured in galactose medium, they become dependent on their mitochondria for ATP synthesis via OXPHOS. Due to this dependence, it has been shown previously that galactose-conditioned cells reveal mitochondrial dysfunction where cells cultured with glucose do not³. Therefore, we assume that the formation of TNTs and the associated rescue mechanism that occurs in galactose-conditioned HeLa cells is due to their dependence on mitochondria for energy production, meaning that the cells cannot cope with the trap-induced import dysfunction. In glucose-cultured cells, presumably they can flourish without fully functional mitochondria due to their highly glycolytic nature, meaning they do not rely on the mitochondria for their energy supply. We have now discussed this in more detail in the text.

Reviewer #2 (Remarks to the Author):

Hi

Thank you by exploring a novel and great area of research, cell to cell communication and amplification of disease. The consequences of this manuscript can provide explanations for multiple diseases that spread cell to cell.

The approach is smart and well done to support the conclusions

Some comments for the reader include

1. The inclusion of quantification of the data specially colocalization and efficiency of the transfer
See response to reviewer 1, point 5.

2. TNT formation time course
See response to reviewer 1, point 4.

3. High magnification of the pictures (insets) can provide a better demonstrations of the points described

Thanks for the suggestion – we have now added insets to imaging panels where appropriate to provide better demonstrations of the points described.

4. How Fig. S3, change respect to time?

Fig. S3 highlights the pixel intensity across the mitochondria after 48 hours of trapping (the same timepoint at which the SIM was carried out; images shown in Fig. 1B). We chose this timepoint as the endpoint as it represents the peak of expression (following transient transfection) and therefore at this time, the cells will be in a sort of equilibrium (the protein of interest has been expressed and either imported into mitochondria or trapped on the outside of mitochondria/ retained in the cytosol). The pixel intensity plots are just a representation of the images shown (to make it more clear and easier for the reader to visualise the protein surrounding mitochondria). However, we presume that the pattern shown in the plots would not change drastically over time.

5. Fig. S4, can be only indicated in the text.

This is indicated in the text – p3 (lines 81-83) – ‘Interestingly, within the same cell, structured illumination microscopy (SIM) showed that some mitochondria (43%) were enveloped by aggregated precursor protein, while others were not (Fig. 1B—cyan and yellow arrows, respectively; S3 and S4).’

6. Fig. 1C and S5 pictures are the same. Please provide new ones

Fig. S5 was an expansion of Fig. 1C, so needs to be the same cell. To avoid confusion, we have merged them into the same figure. Therefore, the whole figure is now shown in Fig. 1C.

7. Can the author provide better pictures (in my copy looks low and high intensity) for S8 and S9.
High-resolution pictures have been included in the resubmission.

8. Is Fig. 1E the same than S12?. Provide a new one. Repeating the same figure is not OK

These figures are different. The traces shown in previous Fig. 1E (now Fig. S7) represent import after chronic/ acute trapping in the absence of any drugs. The trace in previous Fig. S12 (now Fig. S16) represents import after chronic trapping in the presence of nocodazole.

9. Volcano plot in S14 do not have labeled targets

The volcano plot in previous Fig. S14 (now Fig. S18) is a control, as the over-produced protein (that we trapped in the IP) is non-mitochondrial (as it does not have an MTS) and therefore is solely to control for any noise and background effects of treatment with MTX. However, we have now labelled the single significantly increased protein (transportin-1) that is pulled down in the presence of MTX, and this has been clarified in the text.

But overall, this is a great manuscript

Thank you!

References

- 1 Needs, H. I., Lorriman, J. S., Pereira, G. C., Henley, J. M. & Collinson, I. The MitoLuc Assay System for Accurate Real-Time Monitoring of Mitochondrial Protein Import Within Mammalian Cells. *Journal of Molecular Biology* **435**, 168129 (2023).
<https://doi.org/10.1016/j.jmb.2023.168129>
- 2 Ford, H. C. *et al.* Towards a molecular mechanism underlying mitochondrial protein import through the TOM and TIM23 complexes. *Elife* **11** (2022).
<https://doi.org/10.7554/eLife.75426>
- 3 Aguer, C. *et al.* Galactose Enhances Oxidative Metabolism and Reveals Mitochondrial Dysfunction in Human Primary Muscle Cells. *Plos One* **6**, 11 (2011).
<https://doi.org/10.1371/journal.pone.0028536>

REVIEWER COMMENTS

Reviewer #1 (Remarks to the Author):

Comments to the authors

In the revised manuscript, the authors well addressed my previous concerns. Particularly, newly added Fig. 1G convincingly showed that their import block system works properly and their originally developed import reporter is very sensitive tool to monitor mitochondrial import efficiency. At the same time, some critical concerns also arose. Please see specific comments below.

Major points

1. (related to previous comment 1) I agree with that the mitochondrial targeting-efficiency of Su9-mScarlet (or EGFP)-DHFR reduced after MTX treatment in IF in Fig. 1B and Fig. S2. However, the immunoblot in Fig. S1 does not precisely recapitulate this observation. There is no difference in band intensity between lane 3 and 4, and also, there is no increase of precursor in cytosolic fraction after MTX treatment. Therefore, I cannot fully agree with the authors' description of this immunoblot data in the text. The discrepancy might result from the difficulty of separation of the precursor form and mature form on immunoblot as the authors mentioned (although ideally this should be confirmed by knockdown of MPP proteases). I suggest the authors to remove this confusing data Fig. S1 as readers cannot interpret this data properly unless the SDS-PAGE conditions etc are precisely optimized. I think IF image data are convincingly shown and enough to support the authors' conclusion.

2. (related to previous comment 2) To directly answer to my question, the authors are encouraged to examine whether TOM and TIM complexes bind to Su9-EGFP-DHFR before and after MTX treatment. Based on the precise optimization in Fig. 1G, it can be examined after MTX treatment for 2 hours.

3. (related to previous comment 4) The data that monitor the time-dependent change of import efficiency after MTX addition (Fig. 1G) is really convincing and now I fully believe that the authors' precursor trapping system works in mammalian cells, which is good. However, one biggest concern also arises now. That is about the time course of TNT formation (only tested after 48 hours of MTX treatment) and that of the import recovery (it starts from 4 hours as indicated in Fig. 1G). If the TNT formation plays major and critical roles in resolving the import problems as the authors expect (line 219-222 on page 7-8), it should be observed within 4 hours of MTX treatment. This point is very important to prove their main conclusion. It can be also tested through a different experiment – is the import efficiency recovered in HeLa cells that are cultured in glucose medium? Under this culture condition, the TNT formation was

not observed as indicated in Fig. 2B, but MTX-mediated import trapping should be induced. Therefore, the recovery of import (Fig. 1G) or mixed mitochondria (Fig. 3B) would not be observed in HeLa GLU. Although the authors addressed this point by nocodazole treatment (Fig. 4C), nocodazole may induce pleiotropic effects on multiple cellular events, for example, not only inhibiting the TNT formation but also altering mitochondrial integrity. Therefore, the nocodazole experiment is not sufficient to prove their conclusion.

4. (related to previous comment 5) Please describe the experimental conditions of “control” in figure legend. As mentioned above, the co-culture experiment also should be conducted in HeLa cells that are cultured in glucose medium as a precise control. Under these conditions, TNT formation does not happen, so the transfer of mitochondria would not be observed.

5. (related to previous comment 6) I totally understand and fully respect the authors’ trial to address the mechanisms of TNT formation through interactome analysis in Fig. 4D and 4E. However, I’m not sure how informative it is for readers without any experimental verifications. Currently, the analysis was conducted after 48 hours treatment of MTX. But if the TNT formation-mediated import recover happens within 4 hours, the analysis at more earlier time course would be better to address the actual mechanism of TNT formation in this context. However, this needs a lot of work, so I agree with that it is beyond the scope at the current stage.

Minor points

1. In Fig. S2, the labelling of MTX treatment (“– “or “+) should be wrong.

2. As indicated in new Fig. 1G, the time course of MTX treatment is very important factor in this experimental setting. Therefore, the duration of MTX treatment should be clearly indicated in each figure legend, for example in Fig. S9.

Reviewer #2 (Remarks to the Author):

Hi

Thank you for resubmitting this exciting manuscript in the role of TNTs in mitochondrial exchange and protein aggregation.

The manuscript improved but still significant issues remains

1. Give the real numbers with standard deviation
2. There missing figures in the supplemental material
3. The volcano plots are empty
4. The TNT pictures are not convincing. A TNT connects 2 or more cells, some pictures the connection is unclear
5. The authors reused the pictures from the main figures into the supplemental figures. Do you have more pictures that demonstrate the same point?
6. Confocal in the supplemental material is not good quality. Everything is stained.
7. Supplemental Fig. 14, the TNT is unclear, there is not communicating to anything

There are several of the concerns addressed, but the resubmission was not well done.

Reviewer #3 (Remarks to the Author):

I have been recruited as a reviewer on this manuscript after its first round of review to comment on the proteomics data of the study. As such, I will first comment on my overall impressions of the paper, the proteomics data itself, and my perception of how the reviewer comments were addressed.

This is a very interesting study on tunneling nanotube (TNT) formation as a rescue mechanism for mitochondrial protein import clogging in cells. Overall, the data are well presented and support the model. However, I do feel that multiple points have been overstated, and should be modified by the authors before acceptance of this manuscript. (See points below).

In regards to the proteomics data – the data seem to have been collected appropriately, and the presentation in Figure 4E is more or less easy to follow. I do have some suggestions to make these data, as well as those reported in the Supplemental Table, more accessible to the reader. While I appreciate the authors including a negative control experiment to show the specificity of the associations of the target trapping protein, I cannot visualize any dots on this plot but rather only see a single label on the graph. I thus cannot comment on the quality of the negative control experiment.

Finally – I think the authors have done a good job of addressing most of the comments from the reviewers. There are a handful of comments that I believe could be further addressed, which I will outline below.

1. Overstated claims

A. In Figure 1, the authors suggest that precursor stalling triggers enhanced phosphorylation on Drp1 S616. Specifically, they state “There was no significant increase in the ratio of p-DRP1 S616/total DRP1 in the -MTS +MTX condition, highlighting that the significant increase in p-DRP1 S616 observed following precursor trapping (+MTS +MTX) can, at least in part, be attributed to mitochondrial import dysfunction induced by precursor trapping.” (Lines 107-110.) Upon examination of the primary data, it seems that there is absolutely an effect from MTX treatment alone, both on the representative blot in Figure 1D and in the associated quantification. The way this is worded insinuates that the increase in DRP1 phosphorylation is solely due to precursor trapping, which is clearly not true due to this effect of MTX alone. The authors should qualify this in the wording of the manuscript beyond noting that there is a lack of statistical significance, perhaps by stating that “while there is an effect of MTX alone, it fails to reach statistical significance, thus...” . Of note, this was also brought up by Reviewer #1 (see point 3).

B. Lines 157-158: “The fact that this occurrence was dependence on galactose conditioning confirms that the response is mediated by mitochondria”. Change “confirms” to “is consistent with”; there are non-mitochondrial explanations for this phenomenon as well.

C. Lines 212-213: “Overall, we can conclude that the unexpected resilience of import function, within cells subject to precursor trapping, is dependent on TNTs.” No, the data are consistent with a dependence on TNTs, but the rescue with nocadazole, particularly when treated for 48 hrs, does not solely function in the cell to block TNTs. Thus, one cannot draw a direct conclusion here without further experimentation. (To clarify, I am suggesting you change the wording instead of doing more experiments.)

D. Lines 162-163: “In doing so, the number of protrusions was reduced to the baseline levels (Figure 2D, S11), consistent with their identity as TNTs. Figure 2D only shows vehicle v nocadazole treatment in the context of import stalling, thus no baseline condition is shown for comparison. If the authors want to keep this statement regarding a return to “baseline levels”, these data should be included in Figure 2D.

2. Modification of proteomics data

A. In Figure 4E, it is impossible to tell which dots correspond with the labels shown. These should have lines drawn to specifically demonstrate which dot is represented by which protein.

B. Figure 4E lacks an x-axis label; please include.

C. The inclusion of proteomics data as a PDF table in the supplement is not overly user friendly, as it is not easily sortable. It would be better to include this data as an excel spreadsheet including Uniprot IDs (assuming these were captured during analysis) as well as raw intensity values (or Log2 transformed intensity values) to allow analysis of the raw data by readers. Additionally, I believe it would be helpful to include an analysis of mitochondrial versus non-mitochondrial proteins, which could further be commented on regarding mechanistic insights (see more on point 3A. below).

3. Response to reviewer comments

A. Reviewer 1 brought up the point that “mechanistic insights of TNTs formation in this context remained to be speculative.” – a point I agree with. The authors stated in their rebuttal that “We feel that a validation of the TMT proteomic data is beyond the scope of this paper” – I also agree with this. However, I believe the proteomics data, as presented, are not particularly helpful or satisfying to the reader. I believe a few analyses that could be easily done without more experimentation could assist with this. These include:

- i. An analysis of mitochondrial versus non-mitochondrial interactors. Are there more mitochondrial versus non-mitochondrial proteins that interact with the import cloggers? Is it known which sub-compartment they are in? The authors could reference the MitoCoP compendium for such an analysis, which I believe would be relatively straightforward.
- ii. GO term enrichment or some type of analysis to see if there are any pathways that are enriched with the association.

Even if these show no obvious trends, the inclusion of these analyses would be beneficial to then suggest the complexity of the response.

B. In comment #6, Reviewer #1 asked whether ablation of any hits identified in Figure 4E could disrupt the formation of TNTs in this context, and this would be valuable to be followed up on.

POINT BY POINT RESPONSE TO THE REVIEWER COMMENTS VERSION 2:
20th October, 2023

REVIEWER COMMENTS

Reviewer #1 (Remarks to the Author):

Comments to the authors

In the revised manuscript, the authors well addressed my previous concerns. Particularly, newly added Fig. 1G convincingly showed that their import block system works properly and their originally developed import reporter is very sensitive tool to monitor mitochondrial import efficiency. At the same time, some critical concerns also arose. Please see specific comments below.

Major points

1. (related to previous comment 1) I agree with that the mitochondrial targeting-efficiency of Su9-mScarlet (or EGFP)-DHFR reduced after MTX treatment in IF in Fig. 1B and Fig. S2. However, the immunoblot in Fig. S1 does not precisely recapitulate this observation. There is no difference in band intensity between lane 3 and 4, and also, there is no increase of precursor in cytosolic fraction after MTX treatment. Therefore, I cannot fully agree with the authors' description of this immunoblot data in the text. The discrepancy might result from the difficulty of separation of the precursor form and mature form on immunoblot as the authors mentioned (although ideally this should be confirmed by knockdown of MPP proteases). I suggest the authors to remove this confusing data Fig. S1 as readers cannot interpret this data properly unless the SDS-PAGE conditions etc are precisely optimized. I think IF image data are convincingly shown and enough to support the authors' conclusion.

Agree, we have now removed Fig. S1 and corresponding text.

2. (related to previous comment 2) To directly answer to my question, the authors are encouraged to examine whether TOM and TIM complexes bind to Su9-EGFP-DHFR before and after MTX treatment. Based on the precise optimization in Fig. 1G, it can be examined after MTX treatment for 2 hours.

We looked at this before and it didn't work. This was most likely because the interactions are too weak in detergent solution. It did work for the yeast system, but evidently not the mammalian equivalent studied here – probably due to increased instability issues.

3. (related to previous comment 4) The data that monitor the time-dependent change of import efficiency after MTX addition (Fig. 1G) is really convincing and now I fully believe that the authors' precursor trapping system works in mammalian cells, which is good. However, one biggest concern also arises now. That is about the time course of TNT formation (only tested after 48 hours of MTX treatment) and that of the import recovery (it starts from 4 hours as indicated in Fig. 1G). If the TNT formation plays major and critical roles in resolving the import problems as the authors expect (line 219-222 on page 7-8), it should be observed within 4 hours of MTX treatment. This point is very important to prove their main conclusion. It can be also tested through a different experiment – is the import efficiency recovered in HeLa cells that are cultured in glucose medium? Under this culture condition, the TNT formation was not observed as indicated in Fig. 2B, but MTX-mediated import trapping should be induced. Therefore, the recovery of import (Fig. 1G) or mixed mitochondria (Fig. 3B) would not be observed in HeLa GLU. Although the authors addressed this point by nocodazole treatment (Fig. 4C), nocodazole may induce pleiotropic effects on multiple cellular events, for example, not only inhibiting the TNT formation but also altering mitochondrial integrity. Therefore, the nocodazole experiment is not sufficient to prove their conclusion.

The suggested glucose conditioning experiment is a good idea. However, it may not be the best approach. This is because we have not been able to monitor import and import rescue within glycolytic cells (due to lower activity). So without being able to verify mitochondrial rescue, a negative control in these circumstances would be meaningless. Nevertheless, we understand the reviewer's concern about the effects of nocodazole treatment; assuming that by "pleiotropy" they refer to potential multiple/ off-target effects. Therefore, we conducted an additional experiment to control for these effects, including on mitochondrial integrity (new Fig. 4C).

To recap: we initially showed that the chronic trap has no effect on import, which proceeds at the same levels as no trap (Fig. 4C). However, when we add nocodazole we see an impact on import (reduction), which we conclude to be due to a loss of TNT induced mitochondrial rescue. The new experiment tests the effects of nocodazole on precursor import into mitochondria not subject to trapping. The results (Fig. 4C and S15B) demonstrate that import is not directly affected by nocodazole. This shows that at the concentration and conditions we use nocodazole is not affecting mitochondrial integrity (compromised integrity would severely affect mitochondrial import). Thus, we have eliminated the concerns above about nocodazole. So, with that, and as the referee suggests, Fig. 4C now addresses the point made above. A few words have been added to the text to convey our added confidence that the effect of nocodazole supports our conclusions. We also add a note of caution, stating more experiments are needed to understand the precise mechanism.

4. (related to previous comment 5) Please describe the experimental conditions of "control" in figure legend. As mentioned above, the co-culture experiment also should be conducted in HeLa cells that are cultured in glucose medium as a precise control. Under these conditions, TNT formation does not happen, so the transfer of mitochondria would not be observed. Internal control and its description has now been added to amended figure 3B (top panel).

5. (related to previous comment 6) I totally understand and fully respect the authors' trial to address the mechanisms of TNT formation through interactome analysis in Fig. 4D and 4E. However, I'm not sure how informative it is for readers without any experimental verifications. Currently, the analysis was conducted after 48 hours treatment of MTX. But if the TNT formation-mediated import recover happens within 4 hours, the analysis at more earlier time course would be better to address the actual mechanism of TNT formation in this context. However, this needs a lot of work, so I agree with that it is beyond the scope at the current stage. Agree, action is beyond the scope of this MS, but see response to 3rd referee.

Minor points

1. In Fig. S2, the labelling of MTX treatment ("–" or "+") should be wrong. We have now corrected this labelling mistake.

2. As indicated in new Fig. 1G, the time course of MTX treatment is very important factor in this experimental setting. Therefore, the duration of MTX treatment should be clearly indicated in each figure legend, for example in Fig. S9. This has now been added to all relevant figure legends.

Reviewer #2 (Remarks to the Author):

Hi

Thank you for resubmitting this exciting manuscript in the role of TNTs in mitochondrial exchange and protein aggregation.

The manuscript improved but still significant issues remains

1. Give the real numbers with standard deviation

We have now included the precise number plus the standard deviation wherever values are referenced in the text.

2. There missing figures in the supplemental material

We checked this, but could not identify any obvious gaps.

3. The volcano plots are empty

We re-checked the submission where the volcano plots seem to be alright.

4. The TNT pictures are not convincing. A TNT connects 2 or more cells, some pictures the connection is unclear

TNT imaging is notoriously difficult because they are very fragile. We are in the process of collecting better and 3D rendered images, which will be included in our next MS. Watch this space!

5. The authors reused the pictures from the main figures into the supplemental figures. Do you have more pictures that demonstrate the same point?

In some cases images contained in the MS have been duplicated in the associated supplemental figure. This is where the supplemental figures are an expansion of the main figure.

6. Confocal in the supplemental material is not good quality. Everything is stained.

Perhaps the figures have not been converted to PDF faithfully. This will be checked carefully during the next submission and proofing stage.

7. Supplemental Fig. 14, the TNT is unclear, there is not communicating to anything

There are several of the concerns addressed, but the resubmission was not well done.

See note above. In some cases the TNTs will leave the confocal plane. We are working to improve TNT imaging by 3D rendering (for the next MS).

Reviewer #3 (Remarks to the Author):

I have been recruited as a reviewer on this manuscript after its first round of review to comment on the proteomics data of the study. As such, I will first comment on my overall impressions of the paper, the proteomics data itself, and my perception of how the reviewer comments were addressed.

This is a very interesting study on tunneling nanotube (TNT) formation as a rescue mechanism for mitochondrial protein import clogging in cells. Overall, the data are well presented and support the model. However, I do feel that multiple points have been overstated, and should be modified by the authors before acceptance of this manuscript. (See points below).

In regards to the proteomics data – the data seem to have been collected appropriately, and the presentation in Figure 4E is more or less easy to follow. I do have some suggestions to make these data, as well as those reported in the Supplemental Table, more accessible to the reader. While I appreciate the authors including a negative control experiment to show the specificity of the associations of the target trapping protein, I cannot visualize any dots on this plot but rather only see a single label on the graph. I thus cannot comment on the quality of the negative control experiment.

This appears to be fine from our end, but the figure has been re-inserted in case of a technical error during submission.

Finally – I think the authors have done a good job of addressing most of the comments from the reviewers. There are a handful of comments that I believe could be further addressed, which I will outline below.

1. Overstated claims

A. In Figure 1, the authors suggest that precursor stalling triggers enhanced phosphorylation on Drp1 S616. Specifically, they state “There was no significant increase in the ratio of p-DRP1 S616/total DRP1 in the -MTS +MTX condition, highlighting that the significant increase in p-DRP1 S616 observed following precursor trapping (+MTS +MTX) can, at least in part, be attributed to mitochondrial import dysfunction induced by precursor trapping.” (Lines 107-110.) Upon examination of the primary data, it seems that there is absolutely an effect from MTX treatment alone, both on the representative blot in Figure 1D and in the associated quantification. The way this is worded insinuates that the increase in DRP1 phosphorylation is solely due to precursor trapping, which is clearly not true due to this effect of MTX alone. The authors should qualify this in the wording of the manuscript beyond noting that there is a lack of statistical significance, perhaps by stating that “while there is an effect of MTX alone, it fails to reach statistical significance, thus...” . Of note, this was also brought up by Reviewer #1 (see point 3). We have amended the text as suggested.

B. Lines 157-158: “The fact that this occurrence was dependence on galactose conditioning confirms that the response is mediated by mitochondria”. Change “confirms” to “is consistent with”; there are non-mitochondrial explanations for this phenomenon as well. We have amended the text as requested.

C. Lines 212-213: “Overall, we can conclude that the unexpected resilience of import function, within cells subject to precursor trapping, is dependent on TNTs.” No, the data are consistent with a dependence on TNTs, but the rescue with nocadazole, particularly when treated for 48 hrs, does not solely function in the cell to block TNTs. Thus, one cannot draw a direct conclusion here without further experimentation. (To clarify, I am suggesting you change the wording instead of doing more experiments.) We have amended the text as suggested.

D. Lines 162-163: “In doing so, the number of protrusions was reduced to the baseline levels (Figure 2D, S11), consistent with their identity as TNTs. Figure 2D only shows vehicle v nocodazole treatment in the context of import stalling, thus no baseline condition is shown for comparison. If the authors want to keep this statement regarding a return to “baseline levels”, these data should be included in Figure 2D. We have shown the baseline levels (-MTS -MTX) in Fig 2B. We have now cross-referenced this in the text and clarified that this is what we are referring to as baseline levels.

2. Modification of proteomics data

A. In Figure 4E, it is impossible to tell which dots correspond with the labels shown. These should have lines drawn to specifically demonstrate which dot is represented by which protein. We have amended the labelling on this figure as well as on the control figure.

B. Figure 4E lacks an x-axis label; please include. We have rectified this omission.

C. The inclusion of proteomics data as a PDF table in the supplement is not overly user friendly, as it is not easily sortable. It would be better to include this data as an excel spreadsheet including Uniprot IDs (assuming these were captured during analysis) as well as raw intensity values (or Log2 transformed intensity values) to allow analysis of the raw data by readers. Additionally, I believe it would be helpful to include an analysis of mitochondrial versus non-mitochondrial proteins, which could further be commented on regarding mechanistic insights (see more on point 3A. below).

The entire dataset is now included as an Excel spreadsheet (supplementary Table 1) to replace the PDF table in the supplementary materials. This includes the analysis of mitochondrial vs. non-mitochondrial proteins.

3. Response to reviewer comments

A. Reviewer 1 brought up the point that “mechanistic insights of TNTs formation in this context remained to be speculative.” – a point I agree with. The authors stated in their rebuttal that “We feel that a validation of the TMT proteomic data is beyond the scope of this paper” – I also agree with this. However, I believe the proteomics data, as presented, are not particularly helpful or satisfying to the reader. I believe a few analyses that could be easily done without more experimentation could assist with this. These include:

i. An analysis of mitochondrial versus non-mitochondrial interactors. Are there more mitochondrial versus non-mitochondrial proteins that interact with the import cloggers? Is it known which sub-compartment they are in? The authors could reference the MitoCoP compendium for such an analysis, which I believe would be relatively straightforward.

We carried out analysis of mitochondrial proteins using the MitoCOP compendium as a reference. The resulting data is now included in Table S2 and referred to in the text.

ii. GO term enrichment or some type of analysis to see if there are any pathways that are enriched with the association.

We carried out further analysis using the PANTHER classification system to identify any potentially important pathways from our TMT-MS dataset. This is now included in the manuscript as Fig S18 and Table S2.

Even if these show no obvious trends, the inclusion of these analyses would be beneficial to then suggest the complexity of the response.

B. In comment #6, Reviewer #1 asked whether ablation of any hits identified in Figure 4E could disrupt the formation of TNTs in this context, and this would be valuable to be followed up on.

Agree, these experiments are underway.

**POINT BY POINT RESPONSE TO THE REVIEWER COMMENTS VERSION 1:
27th July, 2023**

Reviewer #1 (Remarks to the Author):

Despite the extensive studies on several stress responsive pathways against mitochondrial import stress in yeast, such mechanisms are largely uncharacterized in mammalian cells. In this manuscript, by establishing an artificial clogger-induced mitochondrial import dysfunction model in mammalian cells, the authors showed that the intercellular exchange of healthy mitochondria and dysfunctional mitochondria through tunneling nanotubes (TNTs) maintains the optimal mitochondrial import capacity against the mitochondrial import stress. Their findings are very intriguing, but following points should be considered.

1) Because this manuscript firstly describes the artificial clogger-induced mitochondrial import dysfunction model in mammalian cells, more careful validation of this assay should be conducted.

We have adapted and extensively characterised the MitoLuc assay to monitor mitochondrial protein import in mammalian (HeLa) cells, and this work has now been published in the *Journal of Molecular Biology* ¹. We have referred to this in the text.

2) Although the authors performed an IP-MASS spec analysis of the mitochondrial import clogger (Fig. 4D), the validation of the hit interactors are not conducted. Therefore, mechanistic insights of TNTs formation in this context remained to be speculative. Please see following specific points.

We feel that a validation of the TMT proteomic data is beyond the scope of this paper, but we include this information for speculation purposes. We make this point clear in the discussion. We could remove this data; however, its inclusion could prove useful to fellow researchers, encouraging further interest and follow-up studies about what factors could be involved in the rescue process. Therefore, we suggest that (assuming the data passes a technical review for quality control) it be included in the revised version. See also note below* in response to point 6.

Major points

1. In Fig. S1, the authors showed that mitochondrial targeting of the mature form of Su9-EGFP-DHFR is diminished after MTX addition-induced DHFR folding. Where is the premature (MTS-unprocessed) form? If the MTX addition prevents the import of Su9-EGFP-DHFR, the premature form should appear on the immunoblotting. Do these premature forms reside in TOM-TIM import complex in mitochondria fraction as clogged precursors?

Unfortunately, the mature protein (-MTS) migrates at the same apparent MW as the full-length precursor (+MTS) (Fig. S1; both versions are visible in the cytosolic fraction). This is quite common for mitochondrial and bacterial precursor proteins, especially when the proteins are quite long as in this case (when the mature and precursor proteins' MWs are too close to distinguish).

Therefore, we cannot tell if the import machinery is clogged by the mature or precursor protein. If the trapped substrate is long enough then the MTS can reach the matrix and be processed (as we have previously shown ²; Fig. 1A). So, the TOM-TIM is probably clogged by a mixture of mature and precursor protein, depending on the efficiency of MTS progression when the precursor is trapped. This point has been elaborated in the text.

2. In Fig. 1B, from the SIM image, the authors estimate that some mitochondria (43%) are enveloped by aggregated precursor proteins. However, this is not clearly demonstrated.

Thanks for pointing this out. We have now expanded Fig. 1B to more clearly demonstrate the association of the precursor with some mitochondria and the lack of association with others. We have also described (in the figure legend as well as in more detail in the methods) how the image

analysis was carried out towards the conclusion that 43% of mitochondria are surrounded by precursor; this is also highlighted in Fig. S3 and S4.

It seems majority of the precursor proteins accumulates in the cytosol. In this artificial clogger assay, how much percentage of TOM-TIM complex are actually occupied with aggregated precursor proteins? The aggregated precursor proteins that coated mitochondria in Fig. 1B can be marked with ubiquitin?

As the reviewer mentions, we can see from Western blotting, confocal imaging, and SIM experiments (Fig. S1, S2, 1B, S3 and S4) that in the presence of Su9-EGFP-DHFR +MTX, some of the trapping protein is accumulated at mitochondria, with the remainder in the cytosol. Our investigation of the impact of trapping on import by MitoLuc assays in the presence of Nocodazole (Fig. 4C and S17) highlight a reduction in import of ~33%. When cells were subjected to acute trapping for 10 min (by permeabilisation and exogenous addition of the trapping precursor +MTX; Fig. 1F and S7) we saw a reduction of ~37%. When cells were exposed to trapping for 2 h (by transfection with DNA encoding the trap protein and incubation with MTX for 2 h), there is a reduction in import of ~48% (Fig. 1G and S8). These data are in line with the quantification from SIM, which indicated that ~43% of mitochondria were surrounded by associated precursors (Fig. 1B, S3, and S4). From this, we can conclude that the trap blocks between one-third and one-half of the cell's TOM-TIM complexes. This has now been discussed in the text.

3. In Fig. 1C and 1D, the authors showed that the induction of mitochondrial import cloggers results in the massive mitochondrial fragmentation through Drp1 activation. However, as shown in Fig. S7, phosphorylation of Drp1 seems to be also enhanced by MTX treatment in the absence of clogger.

We agree with the reviewer that phosphorylation of DRP1 at S616 is also enhanced in our control condition (-MTS +MTX) and have reconsidered our quantification strategy for these Western blots to ensure we account for the off-target effects of MTX treatment.

We decided that the best strategy to account for differences in p-DRP1 that are the result of off-target effects attributable to MTX treatment (and not to import defects) would be to work out a ratio of p-DRP1 S616/Total DRP1, where each of the +MTX conditions is normalised to their respective -MTX counterparts. We have also increased the N (biological replicates) for this experiment to further substantiate the results.

The results of this analysis show that, although there is a slight increase in p-DRP1 S616/total DRP1 in the -MTS +MTX condition, this is not significant. However, upon trapping the precursor with the +MTS +MTX condition, there is a significant increase in the phosphorylation of DRP1 at S616. We have now detailed this in the text, and the new quantification is shown in Fig 1E. In addition, we have added the Western blot to the main figure alongside the quantification (Fig 1D).

Are there any specific reasons that the authors only focus on Drp1? For example, OPA1 processing is not accompanied with this mitochondrial fragmentation?

We initially chose to investigate DRP1 as it is a very well-studied and well-characterised fission protein. And, furthermore, phosphorylation at Serine 616 has been well-characterised as a fission-promoting post-translational modification.

However, we agree with the reviewer's comment that it would be interesting to investigate changes in other fission-related proteins in response to precursor trapping. We, therefore, carried out further experiments to investigate the abundance and processing of OPA1, as well as the abundance of MFF and FIS1. We saw no changes in any of these proteins that were not attributable to the off-target effects of MTX treatment, indicating that the increased mitochondrial fragmentation we observed by SIM can be attributed to increased DRP1 phosphorylation at S616. These data are described in the text and shown in Fig. S6.

TMRM or MitoTracker staining seems to be preserved after the clogger induction, so the authors think that these fragmented mitochondria still maintain the mitochondrial membrane potential?

We carried out additional experimentation to investigate the impact of trapping on the membrane potential and respiratory function of the cells. Analysis of TMRM intensity by confocal imaging showed no change in the membrane potential of cells exposed to chronic trapping, and mitochondrial stress tests using a Seahorse respirometer showed that there was also no change in the cells' respiratory function. This is likely due to the TNT-dependent rescue mechanism rescuing the mitochondrial dysfunction by intercellular mitochondrial transfer. The data is now presented in Fig. S9 and has been described in the text.

4. In Fig. 1E, the logic derived from these data (the chronic trapping vs. the acute trapping) is unclear, because the authors used a completely different method to induce import block in the acute trapping. If a rescue mechanism is expected to be operating under the chronic trapping, mitochondria should experience import problem at earlier time point also in this method. Does the short treatment of MTX (the same method utilized in the chronic trapping) block the import in this mitochondrial import monitoring assay?

Good point. We carried out MitoLuc import assays on cells overproducing the trapping precursor with a time course of MTX treatment. We treated the cells without MTX (no trap) or with MTX for 48 h (chronic trap), 4h, 2h, 1h, and 10 min (trapping time course). After 10 min and 1 h of MTX exposure, we did not see any response (presumably this is too short a time to have any effect with trapping in this manner as it relies on the cells producing enough new trap protein in the given timeframe to block import sites). However, when we trapped for 2 h, we saw a significant reduction in import amplitude of ~48%. Then, after 4 h of trapping, the import amplitude was comparable to the no trap control, suggesting that in the latter 2 hours of trapping, the rescue response rescued the cells' import function. This indicates that within a 2-hour period, the cells can sense the import defect and respond to it by forming TNTs and transferring mitochondria. This data is shown in Fig. 1G and S8.

The imaging data also highlights that as well as being fast (see above) the rescue response is also continuous. In addition to the rescue being activated within the first 4 h of exposure to the trap (Fig. 1G and S8), we observe TNTs connecting cells at 48 hours (Fig 2C, 3A, C, D, and E, and movie S1), suggesting that the rescue response is continually happening. This point has now been clarified in the text.

5. In Fig. 3B, as the evidence of the intracellular mitochondrial transfer, the authors showed one representative cell image that contains cloggers from challenged cells and mitochondrial marker proteins from healthy cells. This is an important data to show that the intracellular mitochondrial transfer happens in this context. Therefore, the quantification is necessary. Also, in these cells, mitochondrial morphology is rescued?

We performed correlation analysis to quantify the degree of co-localisation between mitochondrial markers in cells with mitochondrial mixing compared to controls. The data confirmed that, following mitochondrial mixing, mitochondrial markers were co-localised, indicating that mitochondria transferred via TNTs successfully integrated into the mitochondrial network of the recipient cell. This is now described in the text (and methods) and shown in Fig. S12.

6. Like nocodazole, does the ablation of any factor listed as potential interactors of clogger in Fig. 4D disrupt the TNTs formation in this context?

We identify, through proteomics, other factors that could be involved in TNT formation, and present them as potential leads for further analysis. See also note above* (point 2).

7. TNTs formation by mitochondrial clogger seems to be only observed in HeLa cells that are cultured in galactose medium. It might be better to include potential explanations on this point. When HeLa cells are cultured in galactose medium, they become dependent on their mitochondria for ATP synthesis via OXPHOS. Due to this dependence, it has been shown previously that galactose-conditioned cells reveal mitochondrial dysfunction where cells cultured with glucose do

not ³. Therefore, we assume that the formation of TNTs and the associated rescue mechanism that occurs in galactose-conditioned HeLa cells is due to their dependence on mitochondria for energy production, meaning that the cells cannot cope with the trap-induced import dysfunction. In glucose-cultured cells, presumably they can flourish without fully functional mitochondria due to their highly glycolytic nature, meaning they do not rely on the mitochondria for their energy supply. We have now discussed this in more detail in the text.

Reviewer #2 (Remarks to the Author):

Hi

Thank you by exploring a novel and great area of research, cell to cell communication and amplification of disease. The consequences of this manuscript can provide explanations for multiple diseases that spread cell to cell.

The approach is smart and well done to support the conclusions

Some comments for the reader include

1. The inclusion of quantification of the data specially colocalization and efficiency of the transfer
See response to reviewer 1, point 5.

2. TNT formation time course

See response to reviewer 1, point 4.

3. High magnification of the pictures (insets) can provide a better demonstrations of the points described

Thanks for the suggestion – we have now added insets to imaging panels where appropriate to provide better demonstrations of the points described.

4. How Fig. S3, change respect to time?

Fig. S3 highlights the pixel intensity across the mitochondria after 48 hours of trapping (the same timepoint at which the SIM was carried out; images shown in Fig. 1B). We chose this timepoint as the endpoint as it represents the peak of expression (following transient transfection) and therefore at this time, the cells will be in a sort of equilibrium (the protein of interest has been expressed and either imported into mitochondria or trapped on the outside of mitochondria/ retained in the cytosol). The pixel intensity plots are just a representation of the images shown (to make it more clear and easier for the reader to visualise the protein surrounding mitochondria). However, we presume that the pattern shown in the plots would not change drastically over time.

5. Fig. S4, can be only indicated in the text.

This is indicated in the text – p3 (lines 81-83) – ‘Interestingly, within the same cell, structured illumination microscopy (SIM) showed that some mitochondria (43%) were enveloped by aggregated precursor protein, while others were not (Fig. 1B—cyan and yellow arrows, respectively; S3 and S4).’

6. Fig. 1C and S5 pictures are the same. Please provide new ones

Fig. S5 was an expansion of Fig. 1C, so needs to be the same cell. To avoid confusion, we have merged them into the same figure. Therefore, the whole figure is now shown in Fig. 1C.

7. Can the author provide better pictures (in my copy looks low and high intensity) for S8 and S9. High-resolution pictures have been included in the resubmission.

8. Is Fig. 1E the same than S12?. Provide a new one. Repeating the same figure is not OK

These figures are different. The traces shown in previous Fig. 1E (now Fig. S7) represent import after chronic/ acute trapping in the absence of any drugs. The trace in previous Fig. S12 (now Fig. S16) represents import after chronic trapping in the presence of nocodazole.

9. Volcano plot in S14 do not have labeled targets

The volcano plot in previous Fig. S14 (now Fig. S18) is a control, as the over-produced protein (that we trapped in the IP) is non-mitochondrial (as it does not have an MTS) and therefore is solely to control for any noise and background effects of treatment with MTX. However, we have now labelled the single significantly increased protein (transportin-1) that is pulled down in the presence of MTX, and this has been clarified in the text.

But overall, this is a great manuscript

Thank you!

References

- 1 Needs, H. I., Lorriman, J. S., Pereira, G. C., Henley, J. M. & Collinson, I. The MitoLuc Assay System for Accurate Real-Time Monitoring of Mitochondrial Protein Import Within Mammalian Cells. *Journal of Molecular Biology* **435**, 168129 (2023).
<https://doi.org/https://doi.org/10.1016/j.jmb.2023.168129>
- 2 Ford, H. C. *et al.* Towards a molecular mechanism underlying mitochondrial protein import through the TOM and TIM23 complexes. *Elife* **11** (2022).
<https://doi.org:10.7554/eLife.75426>
- 3 Aguer, C. *et al.* Galactose Enhances Oxidative Metabolism and Reveals Mitochondrial Dysfunction in Human Primary Muscle Cells. *Plos One* **6**, 11 (2011).
<https://doi.org:10.1371/journal.pone.0028536>

REVIEWERS' COMMENTS

Reviewer #1 (Remarks to the Author):

Comments to the authors

Overall, the authors well answered to my concerns. Particularly, the authors added a precise negative control to exclude the side effects of nocodazole treatment, which is highly evaluated. To fully confirm their statement (line 224-225, on page 7), I think they should present that TNT formation is indeed observed around 2-4 hours of the MTX treatment. If not, the authors are encouraged to add appropriate explanation for this point.

If the TNT formation was not observed around 2-4 hours, I think the authors should present those data and mention that the first rapid import recovery may not be mediated by TNT but through unknown mechanisms. In that case, the TNT-mediated mitochondria exchange between neighboring cells may maintain the overall mitochondrial health status to ensure the optimal import capacity. Of course, if the TNT formation actually happens around 2-4 hours, not only at 48 hours, I totally agree with the authors' conclusion.

Reviewer #2 (Remarks to the Author):

Hi

The authors provided a partial answers to the previous comments including but not restricted to

- better TNT pictures for Fig. 2. TNT connect 2 or more cells, demonstration that TNTs connect 2 or more cells is essential. long Processes does not mean TNTs
- removing sections without a proper explanation indicate lack of rigor.
- 3D for TNTs can be done for many labs. Yes, it is complicate but achievable
- TNT blockers are required.
- Protrusions in supplemental material are not convincing

The manuscript could be a great contribution, but the answers to the comments were unfortunate

Reviewer #3 (Remarks to the Author):

The authors have addressed all of my comments and requests appropriately. This is a strong manuscript that will have an important impact in the mitochondrial field.

Reviewer #4 (Remarks to the Author):

This interesting manuscript focuses on understanding how cells respond to defects in mitochondrial import machinery, which has deleterious consequences on mitochondrial homeostasis. The authors provide evidence that cells respond to this situation by obtaining mitochondria from other cell types via tunnelling nanotubes (TNTs). I was asked to review this manuscript after a 2nd revision to assess (1) whether the authors have adequately responded to the remaining concerns raised by Reviewer #2 and (2) whether there are technical concerns that compromise the strength of the scientific claims. I will therefore limit the scope of my review to these matters. My expertise is in mitochondria transfer and not mitochondria import machinery or processes, so I cannot comment on the rigor of the latter.

(1) Overall, I feel that the authors did a good job responding to Reviewer 2's concerns, at least in the version that I reviewed. Specific comments on the last point-by-point exchange are below.

(2) The scientific claims regarding mitochondria transfer are reasonably substantiated, but I do think the authors should specifically acknowledge a limitation in the Discussion that their studies mostly relied upon use of mitochondria dyes to track mito transfer. These dyes can be leaky and produce false-positive results if not verified using a dye-independent approach.

Point-by-point responses with Reviewer 2 in Revision 2, and my independent assessments. For each comment, R2.X refers to the last comment in order, "Authors:" provides the authors' full response, and "Me:" provides my independent assessment.

R2.1. Give the real numbers with standard deviation

Authors: We have now included the precise number plus the standard deviation wherever values are referenced in the text.

Me: The authors addressed this point adequately.

R2.2. There missing figures in the supplemental material

Authors: We checked this, but could not identify any obvious gaps.

Me: I did not find any missing figures in the supplemental material, but the authors do not specify the Supplemental figure subpanels in the main text and instead just highlight "S4" etc instead of referencing Fig S4A, S4B, S4C, or S4D. This occurs throughout most of the manuscript. Specific references to each supplemental figure subpanel in the main text would be appreciated and helpful to the reader.

R2.3. The volcano plots are empty

Authors: We re-checked the submission where the volcano plots seem to be alright.

Me: In the version of the main figures and supplemental figures that I reviewed, I do not see any empty volcano plots.

R2.4. The TNT pictures are not convincing. A TNT connects 2 or more cells, some pictures the connection is unclear

Authors: TNT imaging is notoriously difficult because they are very fragile. We are in the process of collecting better and 3D rendered images, which will be included in our next MS. Watch this space!

Me: While I agree with the authors and find the shown examples of TNTs to be high quality and compelling, I also agree with Reviewer 2. It seems sometimes there is no evidence of a TNT. While this could be due to kinetics of TNT formation and deconstruction or the variable focal planes on which TNTs can exist, the authors should temper their conclusions and make room for other plausible mitochondria transfer mechanisms such as release of extracellular vesicles that contain mitochondria and the release of naked mitochondria for capture by the recipient cells.

R2.5. The authors reused the pictures from the main figures into the supplemental figures. Do you have more pictures that demonstrate the same point?

Authors: In some cases images contained in the MS have been duplicated in the associated supplemental figure. This is where the supplemental figures are an expansion of the main figure.

Me: I think it's OK to reuse the same image in the supplemental figure to show the expanded image – this is analogous to showing a cropped Western blot in the main figure to conserve space and then

including the full Western blot image in the supplemental figure. In my mind this is a sign of rigor and transparency.

R2.6. Confocal in the supplemental material is not good quality. Everything is stained.

Authors: Perhaps the figures have not been converted to PDF faithfully. This will be checked carefully during the next submission and proofing stage.

Me: I did not see any major image quality issues in the supplemental material so am not sure what specific issue Reviewer 2 is pointing to.

R2.7. Supplemental Fig. 14, the TNT is unclear, there is not communicating to anything

There are several of the concerns addressed, but the resubmission was not well done.

Authors: See note above. In some cases the TNTs will leave the confocal plane. We are working to improve TNT imaging by 3D rendering (for the next MS).

Me: I tend to agree with the authors on this matter.

-Jon Brestoff

**POINT BY POINT RESPONSE TO THE REVIEWER COMMENTS VERSION 3:
20th December, 2023**

REVIEWERS' COMMENTS

Reviewer #1 (Remarks to the Author):

Comments to the authors

Overall, the authors well answered to my concerns. Particularly, the authors added a precise negative control to exclude the side effects of nocodazole treatment, which is highly evaluated. To fully confirm their statement (line 224-225, on page 7), I think they should present that TNT formation is indeed observed around 2-4 hours of the MTX treatment. If not, the authors are encouraged to add appropriate explanation for this point.

The results now incorporate a very precise description of the data, including the progression of trapping and the rescue process. We now clearly state that TNT formation at this stage is a presumption, due to difficulties in their visualisation at early time points.

If the TNT formation was not observed around 2-4 hours, I think the authors should present those data and mention that the first rapid import recovery may not be mediated by TNT but through unknown mechanisms. In that case, the TNT-mediated mitochondria exchange between neighboring cells may maintain the overall mitochondrial health status to ensure the optimal import capacity. Of course, if the TNT formation actually happens around 2-4 hours, not only at 48 hours, I totally agree with the authors' conclusion.

In the discussion we now acknowledge that other mechanisms for mitochondrial transfer cannot be excluded

Reviewer #2 (Remarks to the Author):

Hi

The authors provided a partial answers to the previous comments including but not restricted to

- better TNT pictures for Fig. 2. TNT connect 2 or more cells, demonstration that TNTs connect 2 or more cells is essential. long Processes does not mean TNTs
- removing sections without a proper explanation indicate lack of rigor.
- 3D for TNTs can be done for many labs. Yes, it is complicate but achievable
- TNT blockers are required.
- Protrusions in supplemental material are not convincing

The manuscript could be a great contribution, but the answers to the comments were unfortunate
The remarks above were not easy to follow so. See response (below) to the Reviewer #4 brought in to assess our response and technical rigour.

Reviewer #3 (Remarks to the Author):

The authors have addressed all of my comments and requests appropriately. This is a strong manuscript that will have an important impact in the mitochondrial field.

Many thanks!

Reviewer #4 (Remarks to the Author):

This interesting manuscript focuses on understanding how cells respond to defects in mitochondrial import machinery, which has deleterious consequences on mitochondrial homeostasis. The authors provide evidence that cells respond to this situation by obtaining mitochondria from other cell types via tunnelling nanotubes (TNTs). I was asked to review this

manuscript after a 2nd revision to assess (1) whether the authors have adequately responded to the remaining concerns raised by Reviewer #2 and (2) whether there are technical concerns that compromise the strength of the scientific claims. I will therefore limit the scope of my review to these matters. My expertise is in mitochondria transfer and not mitochondria import machinery or processes, so I cannot comment on the rigor of the latter.

(1) Overall, I feel that the authors did a good job responding to Reviewer 2's concerns, at least in the version that I reviewed. Specific comments on the last point-by-point exchange are below.

(2) The scientific claims regarding mitochondria transfer are reasonably substantiated, but I do think the authors should specifically acknowledge a limitation in the Discussion that their studies mostly relied upon use of mitochondria dyes to track mito transfer. These dyes can be leaky and produce false-positive results if not verified using a dye-independent approach.

Note that many of the images track mitochondria by the localisation of fluorescent fusion proteins (DsRed, mScarlet, EFGP) not prone to leakage. The small molecule mitotracker is the gold standard for mitochondrial visualisation. If leakage occurs then we would expect false negatives – very faint or invisible mitochondria. It is difficult to imagine how a false positive could arise – the false impression of a mitochondrion in a vacant region of the cell does not seem likely.

Point-by-point responses with Reviewer 2 in Revision 2, and my independent assessments. For each comment, R2.X refers to the last comment in order, "Authors:" provides the authors' full response, and "Me:" provides my independent assessment.

R2.1. Give the real numbers with standard deviation

Authors: We have now included the precise number plus the standard deviation wherever values are referenced in the text.

Me: The authors addressed this point adequately.

R2.2. There missing figures in the supplemental material

Authors: We checked this, but could not identify any obvious gaps.

Me: I did not find any missing figures in the supplemental material, but the authors do not specify the Supplemental figure subpanels in the main text and instead just highlight "S4" etc instead of referencing Fig S4A, S4B, S4C, or S4D. This occurs throughout most of the manuscript. Specific references to each supplemental figure subpanel in the main text would be appreciated and helpful to the reader.

Supplemental figures are now clearly referenced (Supplementary Fig. 1 etc.).

R2.3. The volcano plots are empty

Authors: We re-checked the submission where the volcano plots seem to be alright.

Me: In the version of the main figures and supplemental figures that I reviewed, I do not see any empty volcano plots.

R2.4. The TNT pictures are not convincing. A TNT connects 2 or more cells, some pictures the connection is unclear

Authors: TNT imaging is notoriously difficult because they are very fragile. We are in the process of collecting better and 3D rendered images, which will be included in our next MS. Watch this space!

Me: While I agree with the authors and find the shown examples of TNTs to be high quality and compelling, I also agree with Reviewer 2. It seems sometimes there is no evidence of a TNT. While this could be due to kinetics of TNT formation and deconstruction or the variable focal planes on which TNTs can exist, the authors should temper their conclusions and make room for other plausible mitochondria transfer mechanisms such as release of extracellular vesicles that contain mitochondria and the release of naked mitochondria for capture by the recipient cells.

These possibilities have now been incorporated into the discussion.

R2.5. The authors reused the pictures from the main figures into the supplemental figures. Do you have more pictures that demonstrate the same point?

Authors: In some cases images contained in the MS have been duplicated in the associated supplemental figure. This is where the supplemental figures are an expansion of the main figure.

Me: I think it's OK to reuse the same image in the supplemental figure to show the expanded image – this is analogous to showing a cropped Western blot in the main figure to conserve space and then including the full Western blot image in the supplemental figure. In my mind this is a sign of rigor and transparency.

R2.6. Confocal in the supplemental material is not good quality. Everything is stained.

Authors: Perhaps the figures have not been converted to PDF faithfully. This will be checked carefully during the next submission and proofing stage.

Me: I did not see any major image quality issues in the supplemental material so am not sure what specific issue Reviewer 2 is pointing to.

R2.7. Supplemental Fig. 14, the TNT is unclear, there is not communicating to anything

There are several of the concerns addressed, but the resubmission was not well done.

Authors: See note above. In some cases the TNTs will leave the confocal plane. We are working to improve TNT imaging by 3D rendering (for the next MS).

Me: I tend to agree with the authors on this matter.

-Jon Brestoff

**POINT BY POINT RESPONSE TO THE REVIEWER COMMENTS VERSION 2:
20th October, 2023**

REVIEWER COMMENTS

Reviewer #1 (Remarks to the Author):

Comments to the authors

In the revised manuscript, the authors well addressed my previous concerns. Particularly, newly added Fig. 1G convincingly showed that their import block system works properly and their originally developed import reporter is very sensitive tool to monitor mitochondrial import efficiency. At the same time, some critical concerns also arose. Please see specific comments below.

Major points

1. (related to previous comment 1) I agree with that the mitochondrial targeting -efficiency of Su9-mScarlet (or EGFP)-DHFR reduced after MTX treatment in IF in Fig. 1B and Fig. S2. However, the immunoblot in Fig. S1 does not precisely recapitulate this observation. There is no difference in band intensity between lane 3 and 4, and also, there is no increase of precursor in cytosolic fraction after MTX treatment. Therefore, I cannot fully agree with the authors' description of this immunoblot data in the text. The discrepancy might result from the difficulty of separation of the precursor form and mature form on immunoblot as the authors mentioned (although ideally this should be confirmed by knockdown of MPP proteases). I suggest the authors to remove this confusing data Fig. S1 as readers cannot interpret this data properly unless the SDS-PAGE conditions etc are precisely optimized. I think IF image data are convincingly shown and enough to support the authors' conclusion.

Agree, we have now removed Fig. S1 and corresponding text.

2. (related to previous comment 2) To directly answer to my question, the authors are encouraged to examine whether TOM and TIM complexes bind to Su9-EGFP-DHFR before and after MTX treatment. Based on the precise optimization in Fig. 1G, it can be examined after MTX treatment for 2 hours.

We looked at this before and it didn't work. This was most likely because the interactions are too weak in detergent solution. It did work for the yeast system, but evidently not the mammalian equivalent studied here – probably due to increased instability issues.

3. (related to previous comment 4) The data that monitor the time-dependent change of import efficiency after MTX addition (Fig. 1G) is really convincing and now I fully believe that the authors' precursor trapping system works in mammalian cells, which is good. However, one biggest concern also arises now. That is about the time course of TNT formation (only tested after 48 hours of MTX treatment) and that of the import recovery (it starts from 4 hours as indicated in Fig. 1G). If the TNT formation plays major and critical roles in resolving the import problems as the authors expect (line 219-222 on page 7-8), it should be observed within 4 hours of MTX treatment. This point is very important to prove their main conclusion. It can be also tested through a different experiment – is the import efficiency recovered in HeLa cells that are cultured in glucose medium? Under this culture condition, the TNT formation was not observed as indicated in Fig. 2B, but MTX-mediated import trapping should be induced. Therefore, the recovery of import (Fig. 1G) or mixed mitochondria (Fig. 3B) would not be observed in HeLa GLU. Although the authors addressed this point by nocodazole treatment (Fig. 4C), nocodazole may induce pleiotropic effects on multiple cellular events, for example, not only inhibiting the TNT formation but also altering mitochondrial integrity. Therefore, the nocodazole experiment is not sufficient to prove their conclusion.

The suggested glucose conditioning experiment is a good idea. However, it may not be the best approach. This is because we have not been able to monitor import and import rescue within glycolytic cells (due to lower activity). So without being able to verify mitochondrial rescue, a

negative control in these circumstances would be meaningless. Nevertheless, we understand the reviewer's concern about the effects of nocodazole treatment; assuming that by "pleiotropy" they refer to potential multiple/ off-target effects. Therefore, we conducted an additional experiment to control for these effects, including on mitochondrial integrity (new Fig. 4C).

To recap: we initially showed that the chronic trap has no effect on import, which proceeds at the same levels as no trap (Fig. 4C). However, when we add nocodazole we see an impact on import (reduction), which we conclude to be due to a loss of TNT induced mitochondrial rescue. The new experiment tests the effects of nocodazole on precursor import into mitochondria not subject to trapping. The results (Fig. 4C and S15B) demonstrate that import is not directly affected by nocodazole. This shows that at the concentration and conditions we use nocodazole is not affecting mitochondrial integrity (compromised integrity would severely affect mitochondrial import). Thus, we have eliminated the concerns above about nocodazole. So, with that, and as the referee suggests, Fig. 4C now addresses the point made above. A few words have been added to the text to convey our added confidence that the effect of nocodazole supports our conclusions. We also add a note of caution, stating more experiments are needed to understand the precise mechanism.

4. (related to previous comment 5) Please describe the experimental conditions of "control" in figure legend. As mentioned above, the co-culture experiment also should be conducted in HeLa cells that are cultured in glucose medium as a precise control. Under these conditions, TNT formation does not happen, so the transfer of mitochondria would not be observed.
Internal control and its description has now been added to amended figure 3B (top panel).

5. (related to previous comment 6) I totally understand and fully respect the authors' trial to address the mechanisms of TNT formation through interactome analysis in Fig. 4D and 4E. However, I'm not sure how informative it is for readers without any experimental verifications. Currently, the analysis was conducted after 48 hours treatment of MTX. But if the TNT formation-mediated import recover happens within 4 hours, the analysis at more earlier time course would be better to address the actual mechanism of TNT formation in this context. However, this needs a lot of work, so I agree with that it is beyond the scope at the current stage.
Agree, action is beyond the scope of this MS, but see response to 3rd referee.

Minor points

1. In Fig. S2, the labelling of MTX treatment ("–" or "+") should be wrong.
We have now corrected this labelling mistake.

2. As indicated in new Fig. 1G, the time course of MTX treatment is very important factor in this experimental setting. Therefore, the duration of MTX treatment should be clearly indicated in each figure legend, for example in Fig. S9.
This has now been added to all relevant figure legends.

Reviewer #2 (Remarks to the Author):

Hi

Thank you for resubmitting this exciting manuscript in the role of TNTs in mitochondrial exchange and protein aggregation.

The manuscript improved but still significant issues remain

1. Give the real numbers with standard deviation

We have now included the precise number plus the standard deviation wherever values are referenced in the text.

2. There missing figures in the supplemental material

We checked this, but could not identify any obvious gaps.

3. The volcano plots are empty

We re-checked the submission where the volcano plots seem to be alright.

4. The TNT pictures are not convincing. A TNT connects 2 or more cells, some pictures the connection is unclear

TNT imaging is notoriously difficult because they are very fragile. We are in the process of collecting better and 3D rendered images, which will be included in our next MS. Watch this space!

5. The authors reused the pictures from the main figures into the supplemental figures. Do you have more pictures that demonstrate the same point?

In some cases images contained in the MS have been duplicated in the associated supplemental figure. This is where the supplemental figures are an expansion of the main figure.

6. Confocal in the supplemental material is not good quality. Everything is stained.

Perhaps the figures have not been converted to PDF faithfully. This will be checked carefully during the next submission and proofing stage.

7. Supplemental Fig. 14, the TNT is unclear, there is not communicating to anything

There are several of the concerns addressed, but the resubmission was not well done.

See note above. In some cases the TNTs will leave the confocal plane. We are working to improve TNT imaging by 3D rendering (for the next MS).

Reviewer #3 (Remarks to the Author):

I have been recruited as a reviewer on this manuscript after its first round of review to comment on the proteomics data of the study. As such, I will first comment on my overall impressions of the paper, the proteomics data itself, and my perception of how the reviewer comments were addressed.

This is a very interesting study on tunneling nanotube (TNT) formation as a rescue mechanism for mitochondrial protein import clogging in cells. Overall, the data are well presented and support the model. However, I do feel that multiple points have been overstated, and should be modified by the authors before acceptance of this manuscript. (See points below).

In regards to the proteomics data – the data seem to have been collected appropriately, and the presentation in Figure 4E is more or less easy to follow. I do have some suggestions to make these data, as well as those reported in the Supplemental Table, more accessible to the reader. While I appreciate the authors including a negative control experiment to show the specificity of the associations of the target trapping protein, I cannot visualize any dots on this plot but rather only see a single label on the graph. I thus cannot comment on the quality of the negative control experiment.

This appears to be fine from our end, but the figure has been re-inserted in case of a technical error during submission.

Finally – I think the authors have done a good job of addressing most of the comments from the reviewers. There are a handful of comments that I believe could be further addressed, which I will outline below.

1. Overstated claims

A. In Figure 1, the authors suggest that precursor stalling triggers enhanced phosphorylation on Drp1 S616. Specifically, they state “There was no significant increase in the ratio of p-DRP1 S616/total DRP1 in the -MTS +MTX condition, highlighting that the significant increase in p-DRP1 S616 observed following precursor trapping (+MTS +MTX) can, at least in part, be

attributed to mitochondrial import dysfunction induced by precursor trapping.” (Lines 107-110.) Upon examination of the primary data, it seems that there is absolutely an effect from MTX treatment alone, both on the representative blot in Figure 1D and in the associated quantification. The way this is worded insinuates that the increase in DRP1 phosphorylation is solely due to precursor trapping, which is clearly not true due to this effect of MTX alone. The authors should qualify this in the wording of the manuscript beyond noting that there is a lack of statistical significance, perhaps by stating that “while there is an effect of MTX alone, it fails to reach statistical significance, thus...”. Of note, this was also brought up by Reviewer #1 (see point 3).
We have amended the text as suggested.

B. Lines 157-158: “The fact that this occurrence was dependence on galactose conditioning confirms that the response is mediated by mitochondria”. Change “confirms” to “is consistent with”; there are non-mitochondrial explanations for this phenomenon as well.
We have amended the text as requested.

C. Lines 212-213: “Overall, we can conclude that the unexpected resilience of import function, within cells subject to precursor trapping, is dependent on TNTs.” No, the data are consistent with a dependence on TNTs, but the rescue with nocadazole, particularly when treated for 48 hrs, does not solely function in the cell to block TNTs. Thus, one cannot draw a direct conclusion here without further experimentation. (To clarify, I am suggesting you change the wording instead of doing more experiments.)
We have amended the text as suggested.

D. Lines 162-163: “In doing so, the number of protrusions was reduced to the baseline levels (Figure 2D, S11), consistent with their identity as TNTs. Figure 2D only shows vehicle v nocodazole treatment in the context of import stalling, thus no baseline condition is shown for comparison. If the authors want to keep this statement regarding a return to “baseline levels”, these data should be included in Figure 2D.
We have shown the baseline levels (-MTS -MTX) in Fig 2B. We have now cross-referenced this in the text and clarified that this is what we are referring to as baseline levels.

2. Modification of proteomics data

A. In Figure 4E, it is impossible to tell which dots correspond with the labels shown. These should have lines drawn to specifically demonstrate which dot is represented by which protein.
We have amended the labelling on this figure as well as on the control figure.

B. Figure 4E lacks an x-axis label; please include.
We have rectified this omission.

C. The inclusion of proteomics data as a PDF table in the supplement is not overly user friendly, as it is not easily sortable. It would be better to include this data as an excel spreadsheet including Uniprot IDs (assuming these were captured during analysis) as well as raw intensity values (or Log2 transformed intensity values) to allow analysis of the raw data by readers. Additionally, I believe it would be helpful to include an analysis of mitochondrial versus non-mitochondrial proteins, which could further be commented on regarding mechanistic insights (see more on point 3A. below).

The entire dataset is now included as an Excel spreadsheet (supplementary Table 1) to replace the PDF table in the supplementary materials. This includes the analysis of mitochondrial vs. non-mitochondrial proteins.

3. Response to reviewer comments

A. Reviewer 1 brought up the point that “mechanistic insights of TNTs formation in this context remained to be speculative.” – a point I agree with. The authors stated in their rebuttal that “We feel that a validation of the TMT proteomic data is beyond the scope of this paper” – I also agree with this. However, I believe the proteomics data, as presented, are not particularly helpful or

satisfying to the reader. I believe a few analyses that could be easily done without more experimentation could assist with this. These include:

i. An analysis of mitochondrial versus non-mitochondrial interactors. Are there more mitochondrial versus non-mitochondrial proteins that interact with the import cloggers? Is it known which sub-compartment they are in? The authors could reference the MitoCoP compendium for such an analysis, which I believe would be relatively straightforward.

We carried out analysis of mitochondrial proteins using the MitoCOP compendium as a reference. The resulting data is now included in Table S2 and referred to in the text.

ii. GO term enrichment or some type of analysis to see if there are any pathways that are enriched with the association.

We carried out further analysis using the PANTHER classification system to identify any potentially important pathways from our TMT-MS dataset. This is now included in the manuscript as Fig S18 and Table S2.

Even if these show no obvious trends, the inclusion of these analyses would be beneficial to then suggest the complexity of the response.

B. In comment #6, Reviewer #1 asked whether ablation of any hits identified in Figure 4E could disrupt the formation of TNTs in this context, and this would be valuable to be followed up on.

Agree, these experiments are underway.

**POINT BY POINT RESPONSE TO THE REVIEWER COMMENTS VERSION 1:
27th July, 2023**

Reviewer #1 (Remarks to the Author):

Despite the extensive studies on several stress responsive pathways against mitochondrial import stress in yeast, such mechanisms are largely uncharacterized in mammalian cells. In this manuscript, by establishing an artificial clogger-induced mitochondrial import dysfunction model in mammalian cells, the authors showed that the intercellular exchange of healthy mitochondria and dysfunctional mitochondria through tunneling nanotubes (TNTs) maintains the optimal mitochondrial import capacity against the mitochondrial import stress. Their findings are very intriguing, but following points should be considered.

1) Because this manuscript firstly describes the artificial clogger-induced mitochondrial import dysfunction model in mammalian cells, more careful validation of this assay should be conducted.

We have adapted and extensively characterised the MitoLuc assay to monitor mitochondrial protein import in mammalian (HeLa) cells, and this work has now been published in the *Journal of Molecular Biology*¹. We have referred to this in the text.

2) Although the authors performed an IP-MASS spec analysis of the mitochondrial import clogger (Fig. 4D), the validation of the hit interactors are not conducted. Therefore, mechanistic insights of TNTs formation in this context remained to be speculative. Please see following specific points.

We feel that a validation of the TMT proteomic data is beyond the scope of this paper, but we include this information for speculation purposes. We make this point clear in the discussion. We could remove this data; however, its inclusion could prove useful to fellow researchers, encouraging further interest and follow-up studies about what factors could be involved in the rescue process. Therefore, we suggest that (assuming the data passes a technical review for quality control) it be included in the revised version. See also note below* in response to point 6.

Major points

1. In Fig. S1, the authors showed that mitochondrial targeting of the mature form of Su9-EGFP-DHFR is diminished after MTX addition-induced DHFR folding. Where is the premature (MTS-unprocessed) form? If the MTX addition prevents the import of Su9-EGFP-DHFR, the premature form should appear on the immunoblotting. Do these premature forms reside in TOM-TIM import complex in mitochondria fraction as clogged precursors?

Unfortunately, the mature protein (-MTS) migrates at the same apparent MW as the full-length precursor (+MTS) (Fig. S1; both versions are visible in the cytosolic fraction). This is quite common for mitochondrial and bacterial precursor proteins, especially when the proteins are quite long as in this case (when the mature and precursor proteins' MWs are too close to distinguish).

Therefore, we cannot tell if the import machinery is clogged by the mature or precursor protein. If the trapped substrate is long enough then the MTS can reach the matrix and be processed (as we have previously shown²; Fig. 1A). So, the TOM-TIM is probably clogged by a mixture of mature and precursor protein, depending on the efficiency of MTS progression when the precursor is trapped. This point has been elaborated in the text.

2. In Fig. 1B, from the SIM image, the authors estimate that some mitochondria (43%) are enveloped by aggregated precursor proteins. However, this is not clearly demonstrated.

Thanks for pointing this out. We have now expanded Fig. 1B to more clearly demonstrate the association of the precursor with some mitochondria and the lack of association with others. We have also described (in the figure legend as well as in more detail in the methods) how the image analysis was carried out towards the conclusion that 43% of mitochondria are surrounded by precursor; this is also highlighted in Fig. S3 and S4.

It seems majority of the precursor proteins accumulates in the cytosol. In this artificial clogger assay, how much percentage of TOM-TIM complex are actually occupied with aggregated precursor proteins? The aggregated precursor proteins that coated mitochondria in Fig. 1B can be marked with ubiquitin?

As the reviewer mentions, we can see from Western blotting, confocal imaging, and SIM experiments (Fig. S1, S2, 1B, S3 and S4) that in the presence of Su9-EGFP-DHFR +MTX, some of the trapping protein is accumulated at mitochondria, with the remainder in the cytosol. Our investigation of the impact of trapping on import by MitoLuc assays in the presence of Nocodazole (Fig. 4C and S17) highlight a reduction in import of ~33%. When cells were subjected to acute trapping for 10 min (by permeabilisation and exogenous addition of the trapping precursor +MTX; Fig. 1F and S7) we saw a reduction of ~37%. When cells were exposed to trapping for 2 h (by transfection with DNA encoding the trap protein and incubation with MTX for 2 h), there is a reduction in import of ~48% (Fig. 1G and S8). These data are in line with the quantification from SIM, which indicated that ~43% of mitochondria were surrounded by associated precursors (Fig. 1B, S3, and S4). From this, we can conclude that the trap blocks between one-third and one-half of the cell's TOM-TIM complexes. This has now been discussed in the text.

3. In Fig. 1C and 1D, the authors showed that the induction of mitochondrial import cloggers results in the massive mitochondrial fragmentation through Drp1 activation. However, as shown in Fig. S7, phosphorylation of Drp1 seems to be also enhanced by MTX treatment in the absence of clogger.

We agree with the reviewer that phosphorylation of DRP1 at S616 is also enhanced in our control condition (-MTS +MTX) and have reconsidered our quantification strategy for these Western blots to ensure we account for the off-target effects of MTX treatment.

We decided that the best strategy to account for differences in p-DRP1 that are the result of off-target effects attributable to MTX treatment (and not to import defects) would be to work out a ratio of p-DRP1 S616/Total DRP1, where each of the +MTX conditions is normalised to their respective -MTX counterparts. We have also increased the N (biological replicates) for this experiment to further substantiate the results.

The results of this analysis show that, although there is a slight increase in p-DRP1 S616/total DRP1 in the -MTS +MTX condition, this is not significant. However, upon trapping the precursor with the +MTS +MTX condition, there is a significant increase in the phosphorylation of DRP1 at S616. We have now detailed this in the text, and the new quantification is shown in Fig 1E. In addition, we have added the Western blot to the main figure alongside the quantification (Fig 1D).

Are there any specific reasons that the authors only focus on Drp1? For example, OPA1 processing is not accompanied with this mitochondrial fragmentation?

We initially chose to investigate DRP1 as it is a very well-studied and well-characterised fission protein. And, furthermore, phosphorylation at Serine 616 has been well-characterised as a fission-promoting post-translational modification.

However, we agree with the reviewer's comment that it would be interesting to investigate changes in other fission-related proteins in response to precursor trapping. We, therefore, carried out further experiments to investigate the abundance and processing of OPA1, as well as the abundance of MFF and FIS1. We saw no changes in any of these proteins that were not attributable to the off-target effects of MTX treatment, indicating that the increased mitochondrial fragmentation we observed by SIM can be attributed to increased DRP1 phosphorylation at S616. These data are described in the text and shown in Fig. S6.

TMRM or MitoTracker staining seems to be preserved after the clogger induction, so the authors think that these fragmented mitochondria still maintain the mitochondrial membrane potential?

We carried out additional experimentation to investigate the impact of trapping on the membrane potential and respiratory function of the cells. Analysis of TMRM intensity by confocal imaging showed no change in the membrane potential of cells exposed to chronic trapping, and mitochondrial stress tests using a Seahorse respirometer showed that there was also no change

in the cells' respiratory function. This is likely due to the TNT-dependent rescue mechanism rescuing the mitochondrial dysfunction by intercellular mitochondrial transfer. The data is now presented in Fig. S9 and has been described in the text.

4. In Fig. 1E, the logic derived from these data (the chronic trapping vs. the acute trapping) is unclear, because the authors used a completely different method to induce import block in the acute trapping. If a rescue mechanism is expected to be operating under the chronic trapping, mitochondria should experience import problem at earlier time point also in this method. Does the short treatment of MTX (the same method utilized in the chronic trapping) block the import in this mitochondrial import monitoring assay?

Good point. We carried out MitoLuc import assays on cells overproducing the trapping precursor with a time course of MTX treatment. We treated the cells without MTX (no trap) or with MTX for 48 h (chronic trap), 4h, 2h, 1h, and 10 min (trapping time course). After 10 min and 1 h of MTX exposure, we did not see any response (presumably this is too short a time to have any effect with trapping in this manner as it relies on the cells producing enough new trap protein in the given timeframe to block import sites). However, when we trapped for 2 h, we saw a significant reduction in import amplitude of ~48%. Then, after 4 h of trapping, the import amplitude was comparable to the no trap control, suggesting that in the latter 2 hours of trapping, the rescue response rescued the cells' import function. This indicates that within a 2-hour period, the cells can sense the import defect and respond to it by forming TNTs and transferring mitochondria. This data is shown in Fig. 1G and S8.

The imaging data also highlights that as well as being fast (see above) the rescue response is also continuous. In addition to the rescue being activated within the first 4 h of exposure to the trap (Fig. 1G and S8), we observe TNTs connecting cells at 48 hours (Fig 2C, 3A, C, D, and E, and movie S1), suggesting that the rescue response is continually happening. This point has now been clarified in the text.

5. In Fig. 3B, as the evidence of the intracellular mitochondrial transfer, the authors showed one representative cell image that contains cloggers from challenged cells and mitochondrial marker proteins from healthy cells. This is an important data to show that the intracellular mitochondrial transfer happens in this context. Therefore, the quantification is necessary. Also, in these cells, mitochondrial morphology is rescued?

We performed correlation analysis to quantify the degree of co-localisation between mitochondrial markers in cells with mitochondrial mixing compared to controls. The data confirmed that, following mitochondrial mixing, mitochondrial markers were co-localised, indicating that mitochondria transferred via TNTs successfully integrated into the mitochondrial network of the recipient cell. This is now described in the text (and methods) and shown in Fig. S12.

6. Like nocodazole, does the ablation of any factor listed as potential interactors of clogger in Fig. 4D disrupt the TNTs formation in this context?

We identify, through proteomics, other factors that could be involved in TNT formation, and present them as potential leads for further analysis. See also note above* (point 2).

7. TNTs formation by mitochondrial clogger seems to be only observed in HeLa cells that are cultured in galactose medium. It might be better to include potential explanations on this point. When HeLa cells are cultured in galactose medium, they become dependent on their mitochondria for ATP synthesis via OXPHOS. Due to this dependence, it has been shown previously that galactose-conditioned cells reveal mitochondrial dysfunction where cells cultured with glucose do not³. Therefore, we assume that the formation of TNTs and the associated rescue mechanism that occurs in galactose-conditioned HeLa cells is due to their dependence on mitochondria for energy production, meaning that the cells cannot cope with the trap-induced import dysfunction. In glucose-cultured cells, presumably they can flourish without fully functional mitochondria due to their highly glycolytic nature, meaning they do not rely on the mitochondria for their energy supply. We have now discussed this in more detail in the text.

Reviewer #2 (Remarks to the Author):

Hi

Thank you by exploring a novel and great area of research, cell to cell communication and amplification of disease. The consequences of this manuscript can provide explanations for multiple diseases that spread cell to cell.

The approach is smart and well done to support the conclusions

Some comments for the reader include

1. The inclusion of quantification of the data specially colocalization and efficiency of the transfer
See response to reviewer 1, point 5.

2. TNT formation time course
See response to reviewer 1, point 4.

3. High magnification of the pictures (insets) can provide a better demonstrations of the points described

Thanks for the suggestion – we have now added insets to imaging panels where appropriate to provide better demonstrations of the points described.

4. How Fig. S3, change respect to time?

Fig. S3 highlights the pixel intensity across the mitochondria after 48 hours of trapping (the same timepoint at which the SIM was carried out; images shown in Fig. 1B). We chose this timepoint as the endpoint as it represents the peak of expression (following transient transfection) and therefore at this time, the cells will be in a sort of equilibrium (the protein of interest has been expressed and either imported into mitochondria or trapped on the outside of mitochondria/ retained in the cytosol). The pixel intensity plots are just a representation of the images shown (to make it more clear and easier for the reader to visualise the protein surrounding mitochondria). However, we presume that the pattern shown in the plots would not change drastically over time.

5. Fig. S4, can be only indicated in the text.

This is indicated in the text – p3 (lines 81-83) – ‘Interestingly, within the same cell, structured illumination microscopy (SIM) showed that some mitochondria (43%) were enveloped by aggregated precursor protein, while others were not (Fig. 1B—cyan and yellow arrows, respectively; S3 and S4).’

6. Fig. 1C and S5 pictures are the same. Please provide new ones

Fig. S5 was an expansion of Fig. 1C, so needs to be the same cell. To avoid confusion, we have merged them into the same figure. Therefore, the whole figure is now shown in Fig. 1C.

7. Can the author provide better pictures (in my copy looks low and high intensity) for S8 and S9.
High-resolution pictures have been included in the resubmission.

8. Is Fig. 1E the same than S12?. Provide a new one. Repeating the same figure is not OK

These figures are different. The traces shown in previous Fig. 1E (now Fig. S7) represent import after chronic/ acute trapping in the absence of any drugs. The trace in previous Fig. S12 (now Fig. S16) represents import after chronic trapping in the presence of nocodazole.

9. Volcano plot in S14 do not have labeled targets

The volcano plot in previous Fig. S14 (now Fig. S18) is a control, as the over-produced protein (that we trapped in the IP) is non-mitochondrial (as it does not have an MTS) and therefore is solely to control for any noise and background effects of treatment with MTX. However, we have now labelled the single significantly increased protein (transportin-1) that is pulled down in the presence of MTX, and this has been clarified in the text.

But overall, this is a great manuscript

Thank you!

References

- 1 Needs, H. I., Lorriman, J. S., Pereira, G. C., Henley, J. M. & Collinson, I. The MitoLuc Assay System for Accurate Real-Time Monitoring of Mitochondrial Protein Import Within Mammalian Cells. *Journal of Molecular Biology* **435**, 168129 (2023).
<https://doi.org/10.1016/j.jmb.2023.168129>
- 2 Ford, H. C. *et al.* Towards a molecular mechanism underlying mitochondrial protein import through the TOM and TIM23 complexes. *Elife* **11** (2022).
<https://doi.org/10.7554/eLife.75426>
- 3 Aguer, C. *et al.* Galactose Enhances Oxidative Metabolism and Reveals Mitochondrial Dysfunction in Human Primary Muscle Cells. *Plos One* **6**, 11 (2011).
<https://doi.org/10.1371/journal.pone.0028536>